

# Modeling coupled nitrification-denitrification in soil with an organic hotspot

Jie Zhang[1], Elisabeth Larsen Kolstad[1], Wenxin Zhang[2], Iris Vogeler [1], Søren O. Petersen[1]

[1]Department of Agroecology, iClimate, Aarhus University, Tjele, Denmark
[2]Department of Physical Geography and Ecosystem Science, Lund University, Lund, Sweden

*Correspondence to: Jie Zhang (jiezh@agro.au.dk)*

**Abstract.** The emission of nitrous oxide ($N_2O$) from agricultural soils to the atmosphere is a significant contributor to anthropogenic greenhouse gas emissions. The recycling of organic nitrogen (N) in manure and crop residues may result in spatiotemporal variability of $N_2O$ production and soil efflux which is difficult to capture by process-based models. We

propose a multi-species, reactive transport model to provide detailed insight into the spatiotemporal variability of nitrogen (N) transformations around such $N_2O$ hotspots, which consists of kinetic reactions of soil respiration, nitrification, nitrifier denitrification, and denitrification represented by a system of coupled partial differential equations. The model was tested with results from an incubation experiment at two different soil moisture levels (-30 and -100 hPa, respectively) and was shown to reasonably well reproduce the recorded $N_2O$ and dinitrogen ($N_2$) emissions, and the dynamics of important carbon

(C) and N components in soil. The simulation indicated that the four populations developed in closely connected, but separate layers, with denitrifying bacteria growing within the manure-dominated zone and nitrifying bacteria in the well-aerated soil outside the manure zone and with time also within the manure layer. The modeled $N_2O$ production within the manure zone was greatly enhanced by the combined effect of oxygen deficit, abundant carbon source and supply of nitrogenous substrates. In the wetter soil treatment with a water potential of -30 hPa, diffusive flux of nitrate ($NO_3^-$) across

the manure-soil interface was the main source of $NO_3^-$ for denitrification in the manure zone, while at a soil water potential of -100 hPa, diffusion became less dominant and overtaken by the co-occurrence of nitrification and denitrification in the manure zone. Scenarios were analyzed where diffusive transport of dissolved organic carbon or different mineral N species were switched off, and they showed that the simultaneous diffusion of $NO_3^-$, ammonium ($NH_4^+$), and nitrite ($NO_2^-$) were crucial to simulate the dynamics of N transformations and $N_2O$ emissions in the model. Without considering solute diffusion

in process-based $N_2O$ models, the rapid turnover of C and N associated with organic hotspots can not be accounted for, and it may result in underestimation of $N_2O$ emissions from soil after manure application. The model and its parameters allow for new detailed insights into the interactions between transport and microbial transformations associated with $N_2O$ emissions in heterogeneous soil environments.



## 1. Introduction

Nitrous oxide ($N_2O$) is a long-lived greenhouse gas (LLGHG) that accumulates in the atmosphere, accounting for about 7 % of the radiative forcing by LLGHGs (World Meteorological Organization, 2021). Globally, $N_2O$ emissions increased from 10-12 Tg N yr$^{-1}$ before the industrial era (Davidson, 2009; Syakila and Kroeze, 2011) to an average of ca. 17 Tg N yr$^{-1}$ in the last decade (Thompson et al., 2019). Agriculture is the dominant contributor to this change, with emissions having increased from 0.3-1.0 Tg N yr$^{-1}$ in 1850 to 3.9-5.3 Tg N yr$^{-1}$ in 2010 (Davidson, 2009; Syakila and Kroeze, 2011; Thompson et al.,

2019; Tian et al., 2020). Manure from animal production systems is responsible for as much as 30-50 % of the global $N_2O$ emissions from agriculture (Oenema et al., 2005). The application of manure to arable land is a widely recommended practice to recycle nitrogen (N) and other nutrients for crop production. However, in wet temperate climate it is also a large and highly variable source of $N_2O$ emissions, of which the extent is determined by manure and soil properties, and field management, with liquid manure having the greatest risk for emissions (Charles et al., 2017).

Manure has the potential to stimulate two key biochemical processes governing $N_2O$ emissions, nitrification and denitrification. They are both regulated by multiple factors such as temperature, acidity, and availability of electron donors and acceptors, among which the interactions are highly non-linear and difficult to predict with simple approaches (Tian et al., 2020). Such interactions may create hotspot areas and moments of $N_2O$ emissions at small scales and have implications at landscape scales (Butterbach-Bahl et al., 2013; Groffman et al., 2009; Wagner-Riddle et al., 2020). Short-lived pulses of

$N_2O$ emission can be induced by precipitation if anoxic soil conditions develop owing to impeded oxygen ($O_2$) supply from the atmosphere, provided that mineral nitrogen is present (Christensen et al., 1990a; Sexstone et al., 1985). In agricultural soils, however, manure and crop residues rich in degradable organic matter can also develop anoxic conditions by acting as a temporary sink for $O_2$ leading to local anoxia even in well-drained soil (Christensen et al., 1990b; Kravchenko et al., 2017), and with a temporal stability that allows for microbial growth (Petersen et al., 1992, 1996). Accordingly, high spatial and

temporal variations in nitrification and denitrification activity, and $N_2O$ emissions, have been reported in manure- and plant residue-amended soils (Kravchenko et al., 2017; Petersen et al., 1992; Taghizadeh-Toosi et al., 2021). When modeling nitrification and denitrification activity in soil, it is important to be able to include the effects of such hotspot environments.

Liquid manure (slurry) containing degradable organic carbon (C) and water is particularly prone to create anoxic hotspots upon field application. A part of the slurry infiltrates the surrounding soil in response to the soil water potential gradient

(Olesen et al., 1997a; Petersen et al., 2003), but particulate matter is immobile and suspended organic particles carried with manure liquid may be trapped in the soil matrix. The extent of slurry redistribution depends on the application method, determining the manure-soil contact, and on the water retention properties of manure solids and soil (Petersen et al., 2003). Manure-saturated soil can retain a higher water content than the surrounding bulk soil for a long period of time (Olesen et al., 1997b, 1997a). The elevated water content in conjunction with intensified $O_2$ consumption rates will result in gradients in

the distribution of oxygen (Petersen et al., 1996; Zhu et al., 2015), which have implications for N transformations.



Ammonium ($NH_4^+$) in manure liquid infiltrating the soil will likely be adsorbed to soil particles (Olesen et al., 1997a), and the growth of nitrifying bacteria can therefore be greatly stimulated at short distance from manure-saturated volumes, where $O_2$ and $NH_4^+$ are both non-limiting factors (Petersen et al., 1992). Meanwhile, the lack of oxygen and higher availability of degradable carbon inside the manure-saturated zone can stimulate the activity of heterotrophic denitrifying bacteria provided

that $NO_3^-$ is available. Controlled experiments (e.g., Nielsen et al., 1996; Nielsen and Revsbech, 1994) showed that denitrification rates in active organic hotspots were promoted by $NO_3^-$ from the soil as well as $NO_3^-$ newly produced through nitrification activity, and that coupled nitrification-denitrification around oxic-anoxic interfaces can account for a large proportion of total denitrification in manure-amended soil (Meyer et al., 2002; Nielsen et al., 1996; Nielsen and Revsbech, 1994; Zhu et al., 2015). In a soil without convective water transport, $NH_4^+$ and $NO_3^-$ ions are transported by diffusion only,

and the supply of soil-borne $NO_3^-$ for denitrification in manure hotspots will decline over time (Nielsen et al., 1996; Petersen et al., 1996), whereas the availability of $NO_3^-$ produced *via* nitrification will increase, and diffusion rates between nitrifying and denitrifying niches will be especially high around soil-manure interfaces where steep concentration gradients can develop (Petersen et al., 1992).

A close association between nitrification and denitrification activity greatly complicates the description of N transformations

and $N_2O$ emissions in models. There have been attempts to describe $N_2O$ formation processes in soil at millimeter-scale through modeling and experiments focusing on soil aggregates, where the effective denitrification rate is governed by the physical constraints on the transport of dissolved $O_2$ associated with aggregate size, external $O_2$ content, and soil respiration (Kremen et al., 2005; Schlüter et al., 2018; Smith, 1980). For example, Kremen et al. (2005) found that, with increasing aggregate radius, anaerobic conditions developed inside the aggregates, and $NO_3^-$ availability gradually became the limiting

factor for denitrification. However, the implication of solute diffusion for denitrification in models of soil with macroscale heterogeneity has not been widely studied. There are numerous models to describe nitrification and denitrification processes, and the production of $N_2O$ and dinitrogen ($N_2$). Simplified models (e.g., Conen et al., 2000; Sozanska et al., 2002) are available to use at the field or regional scale, which do not describe solute and gas movement and associated microbial processes. In process-based models (e.g., Jansson and Moon, 2001; Li et al., 2000), a relatively complete suite of

biochemical processes is generally embedded to describe cycles of water, C, and N for target ecosystems. These models often include the transport of C and N species due to convective water flow, but not always the diffusion process driven by concentration gradients of solutes also in the static water phase. DAISY (Hansen et al., 2012) and APSIM (Holzworth et al., 2014) are examples of models that account for diffusion, but both models do not account for the sequence of oxidation and reduction processes of mineral N, nor simulate $O_2$ directly. More commonly, process models include only C and N (or only

$NO_3^-$) solute transport in soil together with water movement without considering the diffusion process. The example models are DayCent (Parton et al., 1998), DNDC (Li et al., 1992), DSSAT (Jones et al., 1998), PaSim (Riedo et al., 1998), and STICS (Brisson et al., 2003), i.e., models which are widely used for simulating biogeochemical cycles and related outputs (Brilli et al., 2017). This can be expected to explain field conditions where water movement (i.e., rainfall and drainage)



controls the distribution of $O_2$, degradable C, and mineral N. However, it can lead to difficulties in reflecting the turnover of
N in soil with organic hotspots where active transport of N species is important for nitrification and denitrification, but uncoupled from water flow as will be case for extended periods after spring fertilization. Besides the solute transport module, spatial extrapolations are in most modeling approaches made using average site parameters for soil moisture and other key drivers, and the interactions among these drivers leading to hotspots are either not included or are represented at insufficient resolution (Groffman et al., 2009), making those models unsuitable to describe organic hotspots in soils.

In this work, we propose a depth- and time-varying model of $N_2O$ emission from an organic hotspot that builds on a system of partial differential equations (PDEs) describing the transformation of several organic or inorganic components of interest. The model presented in this study simultaneously accounts for diffusional transport of gases (i.e., $O_2$, $CO_2$, $N_2O$, and $N_2$) and solutes (i.e., dissolved C and N components) as well as biochemical processes including soil respiration, nitrification, nitrifier denitrification, and denitrification. Our main focus in this study was not to fit the model rigorously to reproduce
observed $N_2O$ emissions and associated components from manure-amended soil, but rather to account for the dynamics of relevant biochemical processes around manure hotspots in soil in a way that is consistent with experimental evidence. We hypothesized that diffusion constraints on substrates and $O_2$ play an important role in regulating microbial activity in general, and denitrification in particular. More specifically, we modeled a one-dimensional laboratory system, with a manure hotspot embedded within a repacked soil core, during a four-week incubation period. We aimed to (1) characterize the multi-species
temporal and spatial dynamics controlling C and N transformations using a modeling approach, and (2) investigate the extent to which solute diffusion is important for simulating $N_2O$ fluxes.

## 2. Methods and materials

### 2.1 Conceptual model and governing equations

The system investigated in the current study was a repacked soil core with a stagnant water phase and constant soil
temperature. To model the transport and reactions of C and N components, the mass conservation equations were employed to account for time- and depth-varying concentrations of multiple components. The fate of individual components in water-filled pores was governed by biochemical reactions and diffusion, while convection-dispersion was assumed to be negligible. Gas diffusion in air, being about four orders of magnitude faster than that in water, was considered to be the only gas transport scheme in the model.

The mass balance equation for every component $\gamma$ in the model can be described by a PDE:

$$\frac{\partial \theta(z,t) C_\gamma(z,t)}{\partial t} = \frac{\partial}{\partial z}\left(D_\gamma(z,t)\frac{\partial C_\gamma(z,t)}{\partial z}\right) + \sum S_\gamma(z,t) \tag{1}$$





where $\gamma$ = dissolved organic carbon (DOC, *aq*), $NO_3^-$ (*aq*), $NO_2^-$ (*aq*), $NH_4^+$ (*aq*), carbon dioxide ($CO_2$, *g*), $O_2$ (*g*), $N_2O$ (*g*), $N_2$ (*g*), aerobic heterotrophs (*s*), ammonia-oxidizing bacteria (*s*), denitrifiers (*s*), and nitrite-oxidizing bacteria (*s*); the subscripts *aq*, *g*, and *s* indicate whether component $\gamma$ is associated with the aqueous, the gaseous, or the solid phase in the model. $\theta$ represents either the water-filled ($\theta_{aq}$) or the air-filled porosity ($\theta_g$) depending on the phase of $C_\gamma$. $D_\gamma$ is the effective diffusion

coefficient for component $\gamma$ (see Sect. S7.1 in the online supplement for details). For the bacterial populations, the diffusion term is excluded in the equation. $S_\gamma$ is the source or sink term of a $\gamma$-component caused by biochemical reactions, which will be depicted in Sect. 2.2. As a component may participate in several reactions, the 12 PDEs for the above components are coupled by the reaction terms and must be solved simultaneously.

For the cationic species $NH_4^+$ in the model, the adsorbed and dissolved forms were assumed to be at equilibrium and follow
the Freundlich model as defined below (Olesen et al., 1999):

$$C_s = K_\mathrm{F}\left(C_w^{\ N}\right) \tag{2}$$

where $C_s$ is the concentration of adsorbed $NH_4^+$, $C_w$ is the concentration of dissolved $NH_4^+$, $K_F$ is the Freundlich distribution coefficient, and $N$ is the dimensionless Freundlich isotherm exponent.

A retardation factor $R_{NH_4^+}$, describing the effect of cation adsorption to soil particles causing a transport delay, was multiplied with the left-hand side of Eq. 1:

$$R_{NH_4^+} = 1 + \left(\frac{\rho_b}{\theta_w}\right)K_\mathrm{d} = 1 + \left(\frac{\rho_b}{\theta_w}\right)K_\mathrm{F}NC_w^{\ N-1} \tag{3}$$

where $\rho_b$ is the soil bulk density and $K_d$ is the distribution coefficient of $NH_4^+$ between soil solids and water.

## 2.2 Reaction processes

As presented in Fig. 1, the mathematical model developed in this study integrated relevant functional groups of microorganisms in a soil-manure system including aerobic heterotrophs, ammonia-oxidizing bacteria (AOB), nitrite-oxidizing bacteria (NOB), and denitrifying bacteria. The model considered the following processes: aerobic respiration,
nitrification, nitrifier denitrification, and denitrification.

In aerobic decomposition, living heterotrophic microorganisms, in the presence of oxygen, feed upon the organic carbon which serves as a source of energy and is respired as $CO_2$:

$$CH_2O + O_2 \rightarrow CO_2 + H_2O \tag{4}$$

$$S_{CO_2+,r} = \rho_b\mu_{CO_2,r}B_{AER}\frac{[C]}{[C]+kM_{C-CO_2}} \times \frac{[O_2]}{[O_2]+kM_{O_2-CO_2}} \tag{5}$$




where $\mu_{CO_2,r}$ (mmol $CO_2$ $g^{-1}$ biomass $d^{-1}$) is the maximum reaction rate regarding microbial biomass, $B_{AER}$ (g biomass $g^{-1}$ dw) is the biomass of aerobic heterotrophs in soil, $\rho_b$ (g $L^{-1}$) is the bulk soil density, and $kM$ (mmol $L^{-1}$) is the half-saturation

constant of substrates. Further, [C] and [$O_2$] represent available concentrations of carbon and oxygen at the reactive sites of the enzyme which will be described in the following. In the subscript of the reaction velocity, $S$, we use the sign "+" or "-" to indicate production or consumption of the component, and we use the letters "r", "nn", "nd", and "dn" to represent the aerobic respiration, nitrification, nitrifier denitrification, and denitrification processes; the same applies to the equations below.

Three pools of organic carbon were considered: immobile C associated with soil organic matter (immobile SOC), immobile C associated with particulate manure solids (manure POC), and DOC. We assumed that only DOC was the substrate for aerobic respiration, and first-order kinetics were used to describe the conversion from immobile SOC and manure POC, respectively, to DOC:

$$S_{SOC-} = \frac{\partial SOC(z,t)}{\partial t} = -\alpha_{SOC}SOC \tag{6}$$

$$S_{POC-} = \frac{\partial POC(z,t)}{\partial t} = -\alpha_{POC}POC \tag{7}$$

where $S_{SOC-}$ and $S_{POC-}$ (g C $g^{-1}$ dw $d^{-1}$) are the rates of SOC and POC conversion to DOC, and $\alpha$ ($d^{-1}$) is the conversion rate.

Other main biochemical reactions in the model, including respiration, nitrification, nitrifier denitrification, and denitrification, were all assumed to follow Michaelis-Menten kinetics. In the nitrification process, the model assumes that $NH_4^+$ is oxidized directly to $NO_2^-$ by AOB, and subsequently to $NO_3^-$ by NOB (Eqs. 8 and 9). This setup thus does not include the intermediates $NH_2OH$ and NO of ammonia oxidation, in contrast to the explicit description found in Chang et al. (2022) and Chen et al. (2019). $N_2O$ is generated as a by-product of incomplete oxidation of $NH_4^+$ to $NO_2^-$ by AOB (Eq. 10,

adapted from Eqs. 1-3 in Chang et al. (2022)). The consumption of $O_2$ by nitrifiers is included in the source/sink terms. AOB is also responsible for nitrifier denitrification where $NO_2^-$ is reduced to $N_2O$ (Eq. 11, adapted from Eqs. 1 and 6 in Chang et al. (2022)). A two-sided effect of $O_2$, both promotion and inhibition (Eq. 15), is included on this pathway wherein $O_2$ is required to support $NH_4^+$ oxidation while also inhibiting the reduction of $NO_2^-$ to $N_2O$ (Wrage et al., 2001).

The governing equations and reaction velocities of nitrification and nitrifier denitrification are:

$$NH_4^+ + 1.5O_2 \rightarrow NO_2^- + 2H^+ + H_2O \tag{8}$$

$$NO_2^- + 0.5O_2 \rightarrow NO_3^- \tag{9}$$

$$2.5NH_4^+ + 2.75O_2 \rightarrow N_2O + 0.5NO_2^- + 3H^+ + 3.5H_2O \tag{10}$$

$$NO_2^- + NH_4^+ + 0.5O_2 \rightarrow N_2O + 2H_2O \tag{11}$$




$$S_{NO_2^-+,nn} = \rho_b \mu_{NO_2^-,nn} B_{AOB} \frac{[NH_4^+]}{[NH_4^+] + kM_{NH_4^+-NO_2^-}} \times \frac{[O_2]}{[O_2] + kM_{O_2-NO_2^-}} \tag{12}$$

$$S_{NO_3^-+,nn} = \rho_b \mu_{NO_3^-,nn} B_{NOB} \frac{[NO_2^-]}{[NO_2^-] + kM_{NO_2^--NO_3^-}} \times \frac{[O_2]}{[O_2] + kM_{O_2-NO_3^-}} \tag{13}$$

$$S_{N_2O+,nn} = \rho_b \mu_{N_2O+,nn} B_{AOB} \frac{[NH_4^+]}{[NH_4^+] + kM_{NH_4^+-N_2O}} \times \frac{[O_2]}{[O_2] + kM_{O_2-N_2O,nn}} \tag{14}$$

$$S_{N_2O+,nd} = \rho_b \mu_{N_2O+,nd} B_{AOB} \frac{[NO_2^-]}{[NO_2^-] + kM_{NO_2^--N_2O}} \times \frac{[NH_4^+]}{[NH_4^+] + kM_{NH_4^+-N_2O}} \times \frac{[O_2]}{[O_2] + kM_{O_2-N_2O,nd}}$$
$$\times \frac{kI_{N_2O}}{[O_2] + kI_{N_2O,nd}} \tag{15}$$

where $\mu$ (mmol g biomass$^{-1}$ d$^{-1}$) is the maximum reaction rate of microbial biomass in individual steps, $B_{AOB}$ and $B_{NOB}$ (g biomass g$^{-1}$ dw) are the microbial biomass of AOB and NOB in soil, and $kI$ (mmol L$^{-1}$) is the inhibition constant of $O_2$.

In the modeling of denitrification, denitrifiers use the carbon source (DOC) to gain energy and reduce $NO_3^-$ stepwise to $NO_2^-$, $N_2O$, and $N_2$ (Eqs. 16-18). The promotion of denitrification by DOC, and the inhibition by $O_2$, is considered in the model. In each modeled step of denitrification, the reduction of nitrogenous oxides is accompanied by anaerobic respiration whereby $CO_2$ is produced. In addition, the possible inhibition by substrates of the activities of nitrifiers and denitrifiers are neglected in the model. Specifically, some ammonia oxidizing bacteria are sensitive to high levels of their substrate, $NH_4^+/NH_3$, and substrate inhibition of nitrification has been observed in agricultural soils (Koper et al., 2010). Denitrification could be inhibited by high concentrations of $NO_3^-$ substrate (Francis and Mankin, 1977), which is attributed to toxicity to denitrifiers in situations where $NO_2^-$ accumulates (Abeling and Seyfried, 1992; Glass et al., 1997).

The governing equations and reaction velocities of denitrification:

$$NO_3^- + 0.5CH_2O \rightarrow NO_2^- + 0.5CO_2 + 0.5H_2O \tag{16}$$

$$2NO_2^- + CH_2O + 2H^+ \rightarrow N_2O + CO_2 + 2H_2O \tag{17}$$

$$N_2O + 0.5CH_2O \rightarrow N_2 + 0.5CO_2 + 0.5H_2O \tag{18}$$

$$S_{NO_2^-+,dn} = \rho_b \mu_{NO_2^-,dn} B_{DEN} \frac{[NO_3^-]}{[NO_3^-] + kM_{NO_3^--NO_2^-}} \times \frac{[C]}{[C] + kM_{C-NO_2^-}} \times \frac{kI_{NO_2^-}}{[O_2] + kI_{NO_2^-,dn}} \tag{19}$$

$$S_{N_2O+,dn} = \rho_b \mu_{N_2O,dn} B_{DEN} \frac{[NO_2^-]}{[NO_2^-] + kM_{NO_2^--N_2O}} \times \frac{[C]}{[C] + kM_{C-N_2O}} \times \frac{kI_{N_2O}}{[O_{2l}] + kI_{N_2O,dn}} \tag{20}$$





$$S_{N_2+,dn} = \rho_b \mu_{N_2,dn} B_{DEN} \frac{[N_2O]}{[N_2O] + kM_{N_2O}} \times \frac{[C]}{[C] + kM_{C-N_2}} \times \frac{kI_{N_2}}{[O_{2l}] + kI_{N_2,dn}} \tag{21}$$

$$S_{CO_2+,dn} = 0.5 S_{NO_2^-+,dn} + S_{N_2O+,dn} + 0.5 S_{N_2+,dn} \tag{22}$$

We assumed that substrate concentrations in the reaction velocity equations represent the available substrates at the reactive sites of enzymes, and that the availability of substrates is affected by the diffusion of substrates through water films, following the work by Davidson et al. (2012). Therefore, since a decline in soil moisture is accompanied by more disconnected water films, the concentrations of dissolved substrate at the site of enzyme reaction will decline and the

available concentration of gaseous reactants would increase. The available concentrations $[C_{aq}]$ in the aqueous phase at the enzyme reaction site are calculated based on a dimensionless diffusivity $D_{aq}$ and water content; and similarly gaseous concentrations, $D_g$, and air content are used for calculating the available concentration $[C_g]$ in the gas phase:

$$[C_{aq}] = C_{aq} \times D_{aq} \times \theta_{aq}^3 \tag{23}$$

$$[C_g] = C_g \times D_g \times \theta_g^{4/3} \tag{24}$$

where $C_{aq}$ and $C_g$ indicate the actual concentrations of dissolved substrates in water and concentrations in air. $D_{aq}$ and $D_g$ are unitless diffusion coefficients of solute in water and air, respectively (Davidson et al., 2012). The value of $D_{aq}$ was

determined by assuming the extreme condition that $[C_{aq}] = C_{aq}$ for saturated soil, i.e. all of the soluble substrate is available at the reaction site under this condition (Papendick and Campbell, 2015). The value of $D_g$ is determined by another assumed extreme condition that all of the gas is available at the reaction site in completely dry soil (Millington, 1959). The effective diffusivity used in Fick's Law, and in Eq. 1, originated from the same sources we used for the dimensionless diffusivities. See the supporting information (Sects. S6.2 and S7.1) for more details.

In the defined model, the biomass of each microbe is composed of two parts: an existing biomass associated with the bulk soil, and new biomass produced during the simulation. The growth of microbial biomass in the model is proportional to the rate of substrate consumption following the Monod equation and the microbial decay following first-order kinetics:

$$\frac{d[B_{AER}]}{dt} = -y_{AER}/\rho_b f_{Cbio} S_{DOC-,r} - a_{AER} B_{AER} \tag{25}$$

$$\frac{d[B_{AOB}]}{dt} = -y_{AOB}/\rho_b f_{Nbio} S_{NH_4^+-,n} - a_{AOB} B_{AOB} \tag{26}$$

$$\frac{d[B_{NOB}]}{dt} = -y_{NOB}/\rho_b f_{Nbio} S_{NO_2^--,n} - a_{NOB} B_{NOB} \tag{27}$$

$$\frac{d[B_{DEN}]}{dt} = -y_{DEN}/\rho_b f_{Cbio} S_{DOC-,dn} - a_{DEN} B_{DEN} \tag{28}$$



where $y$ is the yield coefficient (g C g$^{-1}$ C or g N g$^{-1}$ N), $f_{Cbio}$ (g C g$^{-1}$ biomass) is the C content of microbial biomass, $f_{Nbio}$ (g N g$^{-1}$ biomass) is the N content of the microbial biomass, $S$ (g C L$^{-1}$ soil d$^{-1}$ or g N L$^{-1}$ soil d$^{-1}$) is the rate of substrate consumption, and $a$ (d$^{-1}$) is the microbial decay rate.

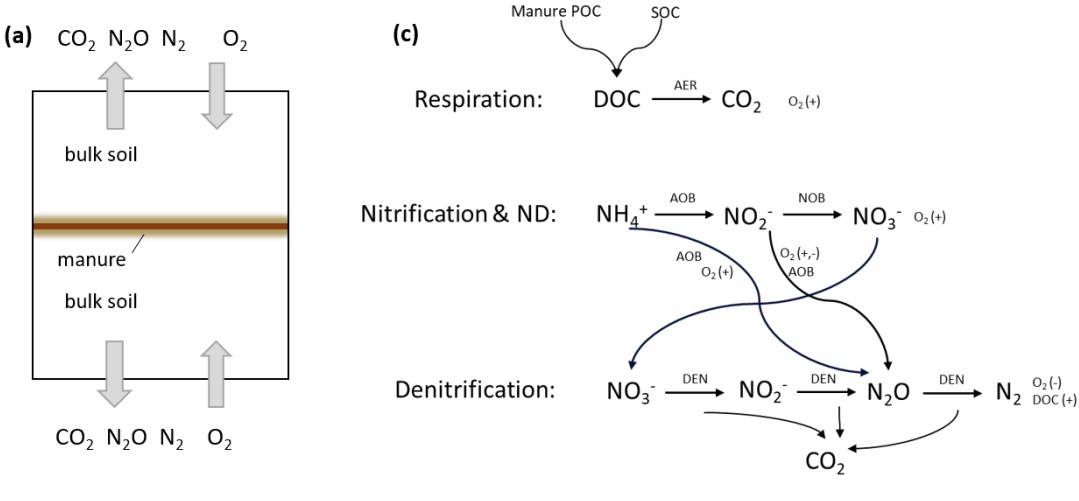

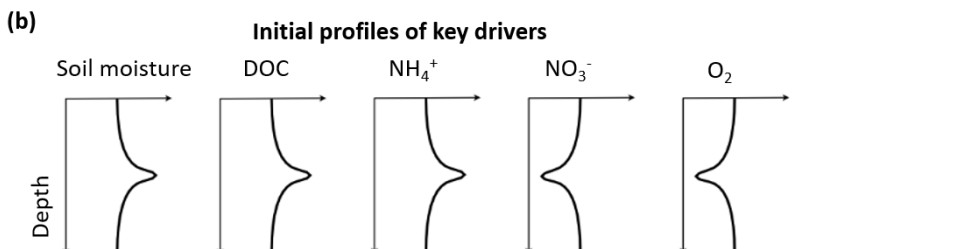

Figure 1: Schematic diagrams of (a) the simulated 10 cm soil core with slurry application and two surfaces connected to the ambient air, (b) conceptualized initial profiles of some key drivers of N cycling in the system, and (c) biochemical processes included in the model. ND: nitrifier denitrification, AER: aerobic heterotrophs, AOB: ammonia-oxidizing bacteria, NOB: nitrite-oxidizing bacteria, and DEN: denitrifiers.

**2.3 Incubation experiment**

The proposed model was first benchmarked against a lab incubation experiment (Kolstad et al., in preparation) investigating the emission of N$_2$O, the total denitrification products, and the spatial distribution of inorganic N after manure incorporation under contrasting levels of soil water content.

In brief, the incubation experiment was set up as follows. Acrylic cylinders (height 5 cm, diameter 8.4 cm) were packed with partially dried and sieved (< 6 mm) sandy loam soil collected in November from the plough layer of a long-term field experiment at Foulumgaard Research Facility, Denmark. On day 0 of the experiment, injection of liquid cattle manure was simulated by applying the manure to the surface of two uniformly packed soil cores which, after the liquid had infiltrated the two cores, were combined with the manure-saturated zone embedded at the center of the now 10 cm high soil cores. The





manure application rate was 0.40 g cm$^{-2}$ (corresponding to 40 t ha$^{-1}$), and basic properties of the manure are listed in Table S1.1. Two soil-manure treatments having a bulk density of 1.4 g cm$^{-3}$ and final soil moisture levels at either -30 hPa or -100 hPa (corresponding to water-filled pore space of 85 % and 70 %, respectively) were used to benchmark the model. For each water potential there was a control treatment, where the soil was packed as described above to the same moisture levels, but slurry was not added. The samples were incubated for 28 days, with gas sampling on day 1, 3, 7, 10, 14, 21, and 28 to

determine fluxes of N$_2$O and CO$_2$ using gas chromatography (GC). Separate samples were sectioned on day 1, 14, and 28 to determine the distribution of NH$_4^+$, NO$_3^-$, NO$_2^-$, and water, while loss on ignition (LOI) was analyzed as a measure of organic matter content on day 1 only. The samples were sliced at 0, 2, 6, 10, 14, 18 and 30 mm distance from the center to either side of the manure layer. The soil NO$_3^-$ pool was labelled with $^{15}$N to allow for both N$_2$O and N$_2$ fluxes to be determined by isotope-ratio mass spectrometry (IRMS); this occurred using extra gas samples collected on day 1, 3, 7, and

220  14.

The experimental results from the two treatments were used for comparison with model outputs, including the daily fluxes of three gases (N$_2$O, N$_2$, and CO$_2$) and measurements of LOI, NH$_4^+$, NO$_3^-$ and NO$_2^-$. The N$_2$ fluxes were calculated from N$_2$O fluxes determined by GC and N$_2$O/N$_2$ ratios determined by IRMS. When N$_2$O or N$_2$ fluxes were below detection limits, N$_2$O/N$_2$ were considered invalid, and these sampling values were then treated as missing data.

**2.4 Model implementation**

**2.4.1 Initial conditions**

The coupled PDEs, as described by Eq. 1 and associated text, had to obey relevant initial and boundary conditions to complete the model setup for the realistic system investigated. The buildup of the soil-manure system started with the application of manure, and redistribution of manure liquid into a larger soil volume until the final water potential in the core

reached equilibrium at -30 or -100 hPa, a process with a duration of a few hours. In conceptualizing the experimental setup in a model system, we took conditions since the slurry redistribution had ceased as the starting point of the model. The soil water distribution was largely constant between day 1 and day 28 (Fig. S1.4) and therefore Eq. 1 where water convection was excluded was suitable to use in the model system.

Given the lack of information about the soil-manure system prior to the first soil sampling on day 1, we made a few

assumptions to set the initial conditions of the model system:

i. Manure liquid was redistributed within a limited "manure zone" and displaced soil pore water resulting in elevated water content compared to before application.

ii. During slurry redistribution, the fate of dissolved components including DOC, NH$_4^+$, and NO$_3^-$ in the slurry was dominated by water flow while biochemical reactions and sorption were neglected, entailing that the components were

uniformly concentrated in the limited zone with a constant total amount.





iii. When slurry redistribution was completed, $O_2$ in the 2 mm center of the limited manure zone, wherein particulate manure solids were concentrated, was in deficit.

The soil core in the model had a depth of 0.1 m from the top (z = 0 m) to the bottom surface (z = 0.1 m), and the center of manure application was at 0.05 m. The vertical extent of the manure zone was determined to be 8 mm, from 0.046 to 0.054

m depth, in which the amount of water on day 1 corresponded to 20.2 g and 17.5 g in the -30 hPa and -100 hPa treatments respectively, close to the 20.9 g of water in the applied slurry. We used the recorded profiles of water content on day 1 as the constant water condition in the modeled soil core.

- Dissolved and particulate C and N

Contributions of DOC, $NH_4^+$, and $NO_3^-$ from both soil and manure were considered in the initial conditions. Soil properties

of control soil on day 1 in the two layers closest to the surfaces, i.e., 0-0.01 m and 0.09-0.1 m, were averaged (Table S1.2). Soil organic carbon (SOC) was estimated from the relationship SOC = 0.39LOI – 0.28 (Jensen et al., 2018). DOC concentrations in the Foulum soil in winter are fairly constant at 20-25 mg C $L^{-1}$ (Gjettermann et al., 2008), and a conversion factor of $3.5 \times 10^{-4}$ between DOC and SOC was used to make estimates of DOC consistent with this reported range. Concentrations of dissolved $NH_4^+$ were calculated from the total content of $NH_4^+$ and the abovementioned Freundlich model.

$NO_3^-$ was assumed to only exist in the water phase. See the supporting information (Sect. S6.3) for details.

Regarding the contributions from the manure, we considered that the total amount of each dissolved component in the applied manure was uniformly distributed in the manure zone of 0.046-0.054 m. Based on the manure application rate and content of volatile solids (VS) in the manure, we estimated that the total organic carbon (TOC) accounted for a fraction of 0.42 of VS (Petersen et al., 2016). We also assumed that the fraction of DOC in TOC was 0.5 (Petersen et al., 1996, 2016),

and hence the amount of DOC in the applied manure was estimated at 30.9 g C $m^{-2}$. Particulate organic carbon (POC) in the manure was then defined as (TOC – DOC); assuming that manure POC was not mobile, this fraction was considered to concentrate within 1 mm from the depth of application, i.e. from 0.049 to 0.051 m depth. Dissolved $NH_4^+$ from manure was estimated from total $NH_4^+$ as above. As there was no detected $NO_3^-$ in the manure, the initial $NO_3^-$ content in the 8 mm manure zone was set to be zero. The initial $NO_2^-$ content was considered to be zero in the entire soil-manure profile. We did

not include the mineralization of organic N from POC in the model as previous studies have associated high C/N with reduced N mineralization in slurry (Barbarika et al., 1985), whereas we considered that mineral N could be immobilized by microbes, i.e., AOB and NOB, and became part of the microbial biomass pool.

The initial content of each component in the 14 sampling layers was thus obtained by summing the contributions of manure and soil.

- Gases





We assumed that the initial concentrations of $CO_2$, $O_2$, $N_2O$, and $N_2$ corresponded to ambient air, i.e., 0.78 atm, 0.21 atm, $4.1 \times 10^{-4}$ atm, and $3.3 \times 10^{-7}$ atm, respectively (World Meteorological Organization, 2021); according to the ideal gas equation, their concentrations in soil air were calculated to be 33.0 mmol $L^{-1}$, 8.9 mmol $L^{-1}$, 0.017 mmol $L^{-1}$, and $1.4 \times 10^{-5}$ mmol $L^{-1}$, respectively. One exception was $O_2$ at the center of the manure zone (i.e., 0.049-0.051 m) where anoxic conditions were assumed.

- Microbial populations

The initial biomass of denitrifiers, AOB, and NOB in the bulk soil were all set to be 12.7 mg $kg^{-1}$ soil referring to the measurement in an arable soil by Khalil et al. (2005) using chloroform fumigation–extraction and microbial enumeration, while the initial biomass of aerobic heterotrophs was assumed to be ten times higher. The new biomass of aerobic heterotrophs, AOB, and NOB initially added with manure were assumed to be zero as we considered the concentrations of these aerobic organisms to be negligible in the anaerobic liquid manure. In contrast, we expected denitrifiers to survive better in the manure, and the denitrifier biomass at the center of the applied manure (0.049-0.051 m) was assumed (including the contribution from soil) to be ten times the base value in soil, and thus similar to the base value of aerobic heterotrophs.

For all of the above components to be solved with Eq. 1, their initial conditions over the modeled soil core were constructed as continuous profiles by interpolating the middle points of all sampling layers (See Figs. S1.1 and S1.2) and extrapolating to the two boundaries.

### 2.4.2 Boundary conditions

For components dissolved in the water phase and microbial populations, we specified no flux at the two boundaries:

$$\frac{\partial C_\gamma(z,t)}{\partial z} = 0, \qquad z = 0 \text{ and } z = 0.1$$

For the concentration of each gaseous component, we specified a constant value for the two boundaries, which was the gas concentration in the ambient air as described in Sect. 2.4.1:

$$C_\gamma(z,t) = C_{atm}(t), \qquad z = 0 \text{ and } z = 0.1$$

### 2.4.3 Computation

The system of coupled equations for eight chemical components and four microbial populations was numerically solved using the partial differential equations solver, *pdepe* in Matlab. The first-order kinetics of SOC and POC were analytically solved and incorporated into the source terms of DOC production to reduce the complexity of the model system. The solver internally uses a dynamic numerical time stepping and the output time step was set to six hours. A non-uniform mesh was applied for the discretization of the 0.1 m one-dimensional model domain. The mesh size was 1 mm in the area away from





the manure-applied zone (z < 0.04 m, z > 0.06 m) and 0.1 mm in the interval of 0.04-0.06 m to obtain stable numerical

solutions at different parameter values in preliminary tests.

For each component, Eq. 1 was valid when the corresponding $\theta$ was above zero. When $\theta$ was equal to zero, there is an inequality between the zero left-hand side and the non-zero right-hand side with the reaction term. In simulating the -30 hPa treatment, a narrow 2 mm saturated zone existed at the center of the core, and to avoid the invalid model domain and not to disturb the rest of model system, a small air fraction of 0.01 was allowed for in the saturated zone, corresponding to a few

drained macropores. Relevant discussion on model uncertainty is included in Sect. 4.3.

The parameter fitting was performed by a combination of algorithm and manual adjustment. We used the surrogate algorithm as the optimizer as it is suitable for evaluating time-consuming models such as the complex PDE system. The objective function used for optimization was the sum of relative root mean squared errors (rRMSEs) of simulated variables from the two treatments:

$$obj = \sum_{j=1}^{N} w_j \frac{\sqrt{\frac{1}{n_j}\sum_{i=1}^{n_j}\left(y_{i,j} - \hat{y}_{i,j}\right)^2}}{\frac{1}{n_j}\sum_{i=1}^{n_j} y_{i,j}} \tag{29}$$


where $obj$ is the value of objective function, $y_{i,j}$ is the $i$-th observation for variable $j$, $\hat{y}_{i,j}$ is the $i$-th simulation for variable $j$, $n_j$ is the number of observations for variable $j$, $w_j$ is the weight of variable $j$, and $N$ is the number of variables.

Parameter values of the C module, $k_{C\_CO2\_r}$, $k_{O2\_CO2\_r}$, and $u_{C\_CO2\_r}$ were optimized by fitting model simulations with measured $CO_2$ fluxes in the -30 hPa and -100 hPa treatments ($w_{CO_2} = 1$ for both treatments), while the rest of parameters

were fixed to the starting values, as we considered the majority of $CO_2$ fluxes to come from aerobic respiration. The values of parameters in the N module were then optimized by fitting the N-related variables with measured data. The N-related variables here included $N_2O$ and $N_2$ fluxes, as well as $NO_3^-$ and $NO_2^-$ content. DOC and $NH_4^+$ were not included as the model-data error was not much affected by parameter fitting in preliminary tests. When presenting the data used to compare with model results in the Results section, we used the labels "measured" and "estimated" to distinguish between the

variables obtained directly from measurements ($CO_2$ fluxes, $N_2O$ fluxes, $NO_3^-$ content, and $NO_2^-$ content, where the latter two were considered to only exist in dissolved form) and the variables ($N_2$ fluxes, DOC, and dissolved $NH_4^+$) estimated from relevant experimental data and assumptions stated earlier.

To ensure good agreements with the temporal and spatial dynamics of multiple variables in the experiment, the selection of precise and accurate weights in objective functions was often difficult as small perturbations of the weights can lead to quite

different solutions (Konak et al., 2006). By trial and error, we assigned the weights of 0.2, 0.05, 0.012, and 0.001 to the



rRMSEs regarding $N_2O$, $N_2$, $NO_2^-$, and $NO_3^-$ in the -30 hPa treatment; and weights of 0.05, 0.05, 0.001, and 0.001 to the corresponding rRMSEs in the -100 hPa treatment. To illustrate, the $N_2O$ peak in the -30 hPa treatment is of high interest in the study and we also added one more term of $N_2O$ peak value error, $(peak_{obs} - peak_{sim})/peak_{obs}$, to ensure the interest was met. Model errors for simulated mineral N were prone to be much higher than for gas emissions and we therefore

assigned them smaller weights to avoid overfitting mineral N accompanied by poor prediction of gas fluxes while minimizing the objective function. After checking the optimized model outputs, which in general aligned well with the measurements, we manually adjusted two parameters ($\mu_{NO2\_dn}$ and $\mu_{N2O\_nd}$) by trials to make the temporal trend of $NO_2^-$ profiles more consistent with the measurements. Model input parameters are described in Sect. S7 in the supporting information.

**2.4.4 Flux calculation**

The efflux of gases from the soil cores was calculated according to Fick's first law, where the concentration gradient between the ambient air and the outmost soil layers at the top and bottom was the driving force of gas flux:

$$J(t) = \frac{D_{\text{eff,g}}(z = z_1, t)}{\Delta z}(C(z = z_1, t) - C_{\text{atm}}) + \frac{D_{\text{eff,g}}(z = z_2, t)}{\Delta z}(C(z = z_2, t) - C_{\text{atm}}) \qquad (30)$$

where $J$ is the gas fluxes (mol m$^{-2}$ d$^{-1}$), $D_{eff,g}$ (m$^2$ d$^{-1}$) is the effective gas diffusion coefficients at the two depths closest to the borders in the discretization of the model where $z_1 = 0.001$ m and $z_2 = 0.099$ m, and $\Delta z$ is the depth interval of 0.001 m.

**2.4.5 Scenario tests**

We took the original setup of the model where the diffusion process of all the solutes was included as the baseline scenario. Based on the calibrated model, we simulated four scenarios in which solute diffusion in the soil-manure system was eliminated to a certain degree. Diffusive fluxes of different solutes were selectively turned off in Scenarios 1-4 as shown in Table 1, where Scenario 1 did not allow any solute diffusion, Scenario 2 allowed only $NO_3^-$ diffusion, Scenario 3 allowed

only $NH_4^+$ diffusion, and Scenario 4 allowed diffusion of $NH_4^+$, $NO_3^-$ and $NO_2^-$ but not DOC. In Eq. 1, turning off the diffusive flux of component $\gamma$ meant that the effective diffusion coefficient $D_\gamma$ was set to zero.

**Table 1: Tested scenarios on the influence of solute diffusion**

|  | Diffusional fluxes turned on (1) or turned off (0) | | | | |
|---|---|---|---|---|---|
|  | Baseline | Scenario 1 | Scenario 2 | Scenario 3 | Scenario 4 |
| DOC | 1 | 0 | 0 | 0 | 0 |
| $NH_4^+$ | 1 | 0 | 0 | 1 | 1 |
| $NO_3^-$ | 1 | 0 | 1 | 0 | 1 |





| NO$_2^-$ | 1 | 0 | 0 | 0 | 1 |
|---|---|---|---|---|---|

## 3. Results

### 3.1 Testing model predictive capability with experimental data

Emissions of three gases, i.e., $N_2O$, $N_2$, and $CO_2$, during four weeks in the experiment were compared with model simulations (Fig. 2). In the -30 hPa treatment with manure application, the model showed a peak $N_2O$ flux on day 3 which was comparable with the peak value of 2880 µg N m$^{-2}$ h$^{-1}$ observed on day 1, but with a delay of two days. In the -100 hPa treatment, the simulated $N_2O$ fluxes in the first three days were between 500 and 1500 µg N m$^{-2}$ h$^{-1}$, which was much higher than the recorded fluxes with a maximum of 130 µg N m$^{-2}$ h$^{-1}$. After the first week the simulation results corresponded well

with the tail of fluxes recorded in the experiment, and small negative values were obtained in the model for the -100 hPa treatment. The simulated $N_2$ fluxes agreed with the magnitude of $N_2$ fluxes estimated from [15]N labeling in the experiment and were approximately one order of magnitude higher than the observed $N_2O$ fluxes in both treatments. Specifically, the temporal trend of $N_2$ fluxes in the -30 hPa treatment was simulated well despite the limited experimental data available. The simulated $CO_2$ fluxes were in general comparable to those observed, especially in the -100 hPa treatment with an early $CO_2$

peak around day 1 in both model and experiment. In the experiment, $CO_2$ fluxes in -30 hPa treatment were generally lower than in the -100 hPa treatment, and model results showed the same pattern. A secondary increase in $CO_2$ flux, observed during the second week in the -30 hPa treatment, was not shown in the simulation.





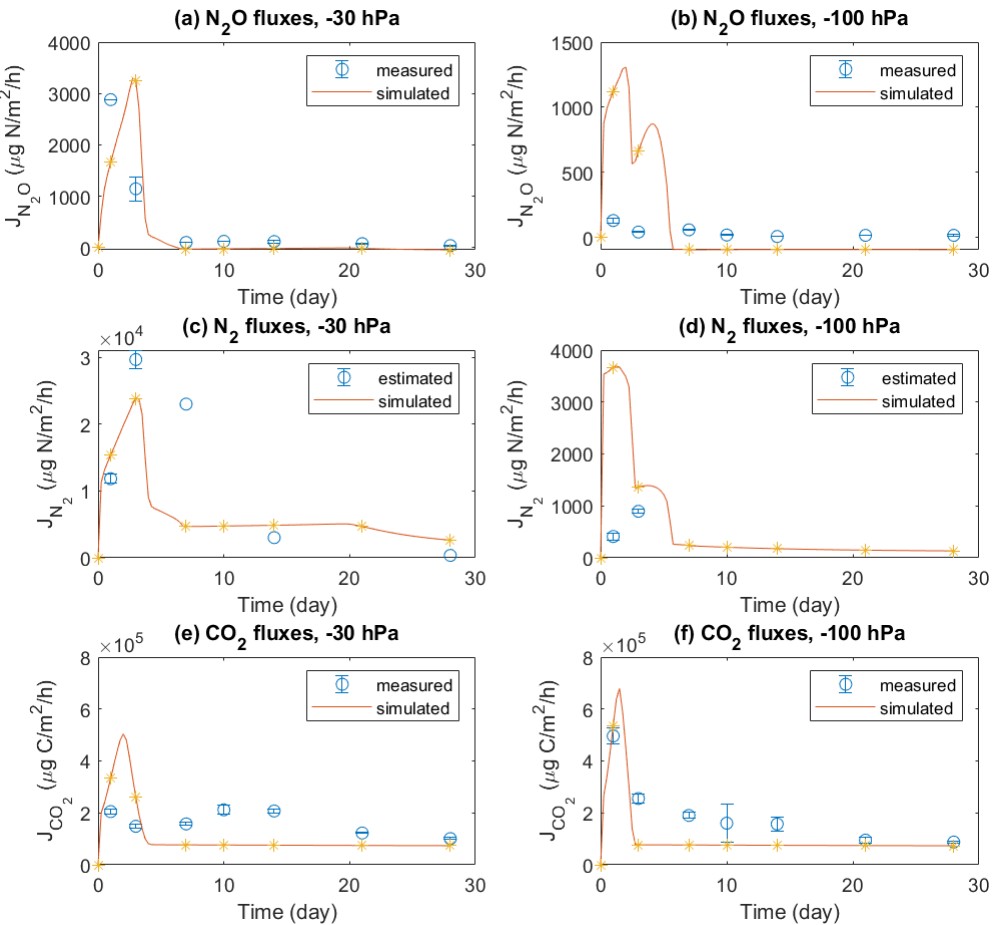

**Figure 2: Measured and modeled N₂O, N₂ and CO₂ fluxes in the -30 hPa and -100 hPa treatments. The N₂ fluxes used for comparison with model results were estimated from the observed N₂O fluxes and $^{15}$N-labeled N₂O/N₂ ratios.**

Mineral N and DOC concentrations in the soil profile were also compared with model simulations (Figs. 3 and 4). The dissolved $NH_4^+$ in Fig. 3a and Fig. 4a, used to compare with model results, was estimated according to a Freundlich adsorption isotherm as described in Sect. 2.4.1. In the manure zone, the estimated concentrations of dissolved $NH_4^+$ by day 1 had dropped from the initial value of 23.1 mmol $L^{-1}$ to 4.1 mmol $L^{-1}$ in the -30 hPa treatment, and from 24.1 mmol $L^{-1}$ to 4.8 mmol $L^{-1}$ in the -100 hPa treatment. The total $NH_4^+$ content in the manure zone was calculated to be 435 mg N $kg^{-1}$ initially if distributed within the 8 mm layer, which was approximately four times the content actually observed on day 1 of 112 and 123 mg N $kg^{-1}$ in the -30 hPa and -100 hPa treatments, respectively (Fig. S2.3a and Fig. S2.4a). In the model, neither the dissolved nor the total $NH_4^+$ content decreased as much as observed in the experiment by day 1, and it took four days before the majority of $NH_4^+$ was reduced to the level observed by day 1 in the experiment. By day 14, the dissolved $NH_4^+$ content in the manure hotspot was reduced to the level in the bulk soil both in the experiment and in the model.





In the experiment the $NO_3^-$ content in the manure zone was negligible one day after manure addition in both treatments. In the -30 hPa treatment, $NO_3^-$ content then increased and approached the background level in the unamended control soil after four weeks. In the simulation, $NO_3^-$ content was comparable with the measurements on day 1 and day 14, but showed a net removal of $NO_3^-$ in the vicinity of the manure zone on day 28, which was not observed in the experiment. In the -100 hPa

treatment of the experiment, $NO_3^-$ content showed an overall increase within the soil profile after two and four weeks compared to day 1 (Fig. 4c). The $NO_3^-$ content on day 28 in the soil outside the manure zone was relatively higher than that on day 14, whereas the $NO_3^-$ content at the center of the manure zone was lower by day 28 compared to day 14, and this trend was well captured by the model (Fig. 4d).

The DOC content was estimated from measured values of LOI and manure VS for comparison with model results. The

estimated DOC profile on day 1 showed a total amount of 20.9 and 23.9 mmol in the -30 hPa and -100 hPa treatments, respectively. The modeled DOC by day 1 showed comparable total amounts of 27.4 and 26.8 mmol in the -30 hPa and -100 hPa treatments, respectively, but with a wider distribution in the soil than estimated in the experiment (Fig. 3e-f and Fig. 4e-f). According to the model simulation, the DOC in the manure zone was mostly removed within two weeks in both treatments, but no further organic matter analyses were done in the experiment to confirm this.

For both treatments the modeled $NO_2^-$ content showed a similar temporal trend as the experimental data. The total $NO_2^-$ content in the soil was higher on day 1 than after 14 and 28 days of incubation (Fig. 3g-h and Fig. 4g-h). The simulation values were generally larger than those recorded, but considering the analytical uncertainty of the low $NO_2^-$ levels in the experiment, this discrepancy was not unacceptable. In contrast to the bell-shaped observed $NO_2^-$ profile on day 1, the simulated $NO_2^-$ profile showed a trough at the center of the manure-saturated zone where $NO_2^-$ concentration developed

more in the following days.





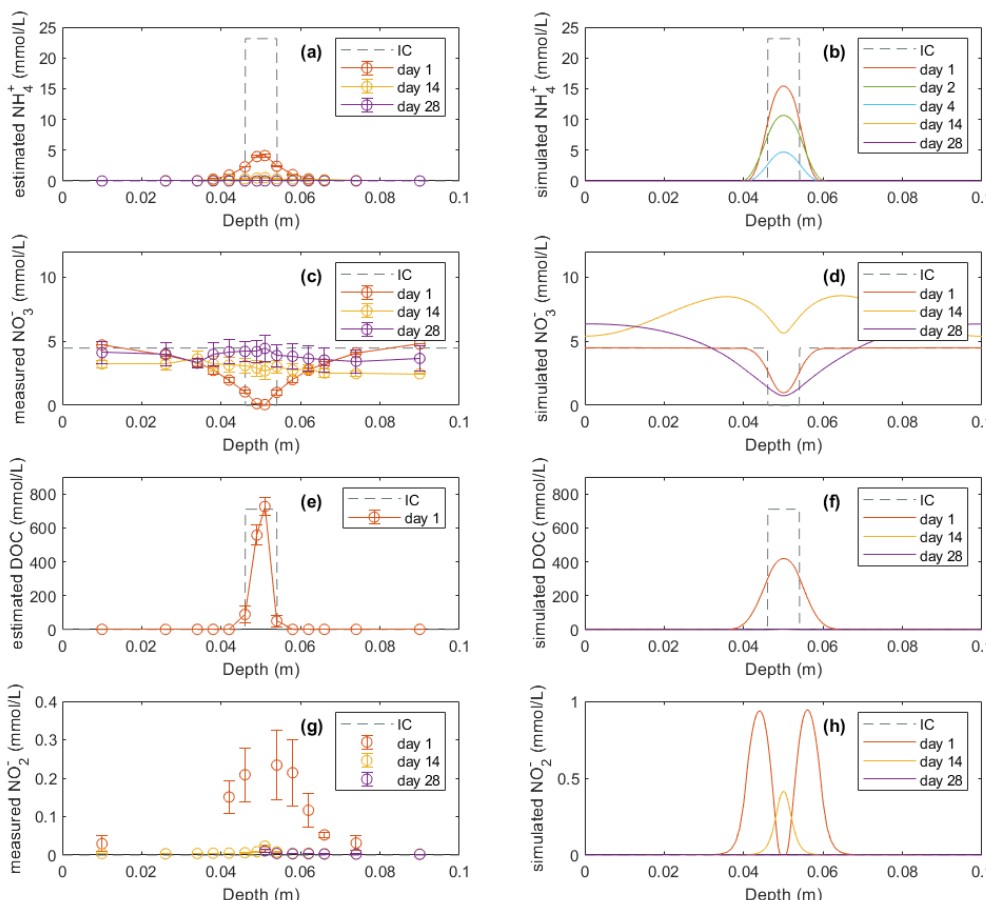

**Figure 3: Model results (right) of soil variables (DOC and dissolved N species) compared with direct measurements or estimations (left) in the -30 hPa treatment. In the legend, IC indicates the initial condition applied in the model to represent the lab experiment with slurry application.**





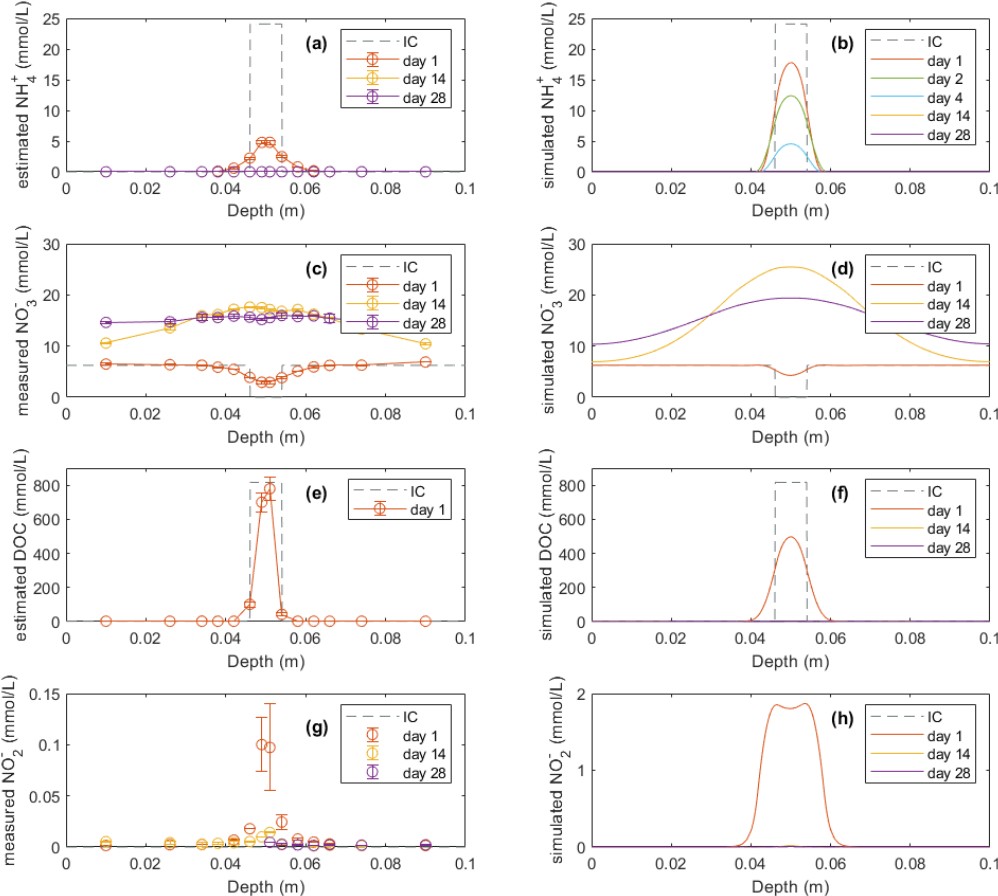

**Figure 4: Model results (right) of soil variables (DOC and dissolved N species) compared with direct measurements or estimations (left) in the -100 hPa treatment. In the legend, IC indicates the initial condition applied in the model to represent the lab experiment with slurry application.**

### 3.2 Spatiotemporal features of simulated N₂O and other variables

Figure 5 shows the modeled process rates related to $N_2O$ transformations in the -30 hPa treatment, including $N_2O$ production *via* nitrification, nitrifier denitrification, and denitrification; $N_2O$ reduction *via* denitrification; and rates of $N_2O$ change *via* diffusion. According to the model, $N_2O$ production mainly occurred at the depth interval 0.04-0.06 m encompassing the manure zone from 0.046 to 0.054 m (Fig. 5a-d). At the center of the manure zone, around the 2 mm saturated area, denitrification was the major pathway of $N_2O$ production, with the highest rate of 6 mmol $N_2O$ $L^{-1}$ soil $d^{-1}$ on day 3 (Fig. 5d). Nitrification contributed to $N_2O$ production in the zone outside the 2 mm saturated area and peaked around the soil-manure interface (Fig. 5b). $N_2O$ production *via* nitrification mainly occurred during the first week, and the reaction rate up to 0.07 mmol $N_2O$ $L^{-1}$ soil $d^{-1}$ were markedly lower than the simultaneous denitrification rates according to the model. The modeled nitrifier denitrification took place close to the saturated zone where the reaction rates were much higher than in other layers





(Fig. 5c). Similar to nitrification and denitrification, N$_2$O production from nitrifier denitrification mainly occurred during the first week, and the highest rate was approx. 0.1 mmol N$_2$O L$^{-1}$ soil d$^{-1}$ on day 3.

Microbial reduction of N$_2$O to N$_2$ showed a similar spatiotemporal pattern as N$_2$O production *via* denitrification (Fig. 5e). In accordance with this, the net reaction, i.e., the sum of production and consumption rates, peaked at the center of the manure zone and had a moderate intensity in the soil volumes mainly associated with nitrification (Fig. 5a). Negative rates around the central manure-saturated zone were paralleled by N$_2$ production (Fig. 5a).

The rates in Fig. 5f, i.e., the first term on the right-hand side of Eq. 1, indicate the rate of N$_2$O concentration change caused by gas diffusion. The negative values in the soil profile represent the decrease in N$_2$O concentrations owing to efflux to the surrounding soil which mainly occurred in the manure zone. Positive values represent a gain in N$_2$O concentrations due to gas diffusion and mainly occurred in the bulk soil where the generated N$_2$O accumulated before escaping to the atmosphere. The net change of N$_2$O concentrations in the soil showed an overall increase on day 1, and then a decrease on day 3 that was followed by minor changes on the following sampling days according to the simulation (Fig. 5g). The direction and magnitude of the internal gas transport of N$_2$O is illustrated in Fig. 5h.

In the -100 hPa treatment, the intensities of N$_2$O production from the three processes were different from the -30 hPa treatment, whereas the spatial distributions of the individual processes were similar to the patterns observed in the -30 hPa treatment (Fig. S3.1a-d). Nitrification rates were generally higher than those of the -30 hPa treatment, especially at the center of the manure zone within the first three days (Fig. S3.1b and Fig. 5b). Specifically, in the -30 hPa treatment the nitrification rate in the central manure-saturated zone was approx. 0.02 mmol N$_2$O L$^{-1}$ soil d$^{-1}$ between day 1 and day 3, which was ca. 50 % lower than the rates at the manure-soil interface (Fig. 5b). In the -100 hPa treatment, the corresponding rate at the manure center increased from 0.04 to 0.06 mmol N$_2$O L$^{-1}$ soil d$^{-1}$ between day 1 and day 3 in the model simulation, and a higher rate was also found at the manure-soil interface (Fig. S3.1b). Denitrification and nitrifier denitrification were less intensive in the -100 hPa treatment compared to the -30 hPa treatment. N$_2$O produced from nitrifier denitrification was up to 0.04 mmol N$_2$O L$^{-1}$ soil d$^{-1}$ on day 3 in the -100 hPa treatment compared with a rate of 0.1 mmol N$_2$O L$^{-1}$ soil d$^{-1}$ in the -30 hPa treatment (Fig. S3.1c and Fig. 5c). Denitrification-derived N$_2$O showed the highest rate with up to 0.8 mmol N$_2$O L$^{-1}$ soil d$^{-1}$ on day 1, markedly lower than the peak value of ca. 6 mmol N$_2$O L$^{-1}$ soil d$^{-1}$ on day 3 in the -30 hPa treatment (Fig. S3.1d and Fig. 5d). The net changes of N$_2$O concentrations showed an increase during the first three days, and slight changes afterwards (Fig. S3.1g).



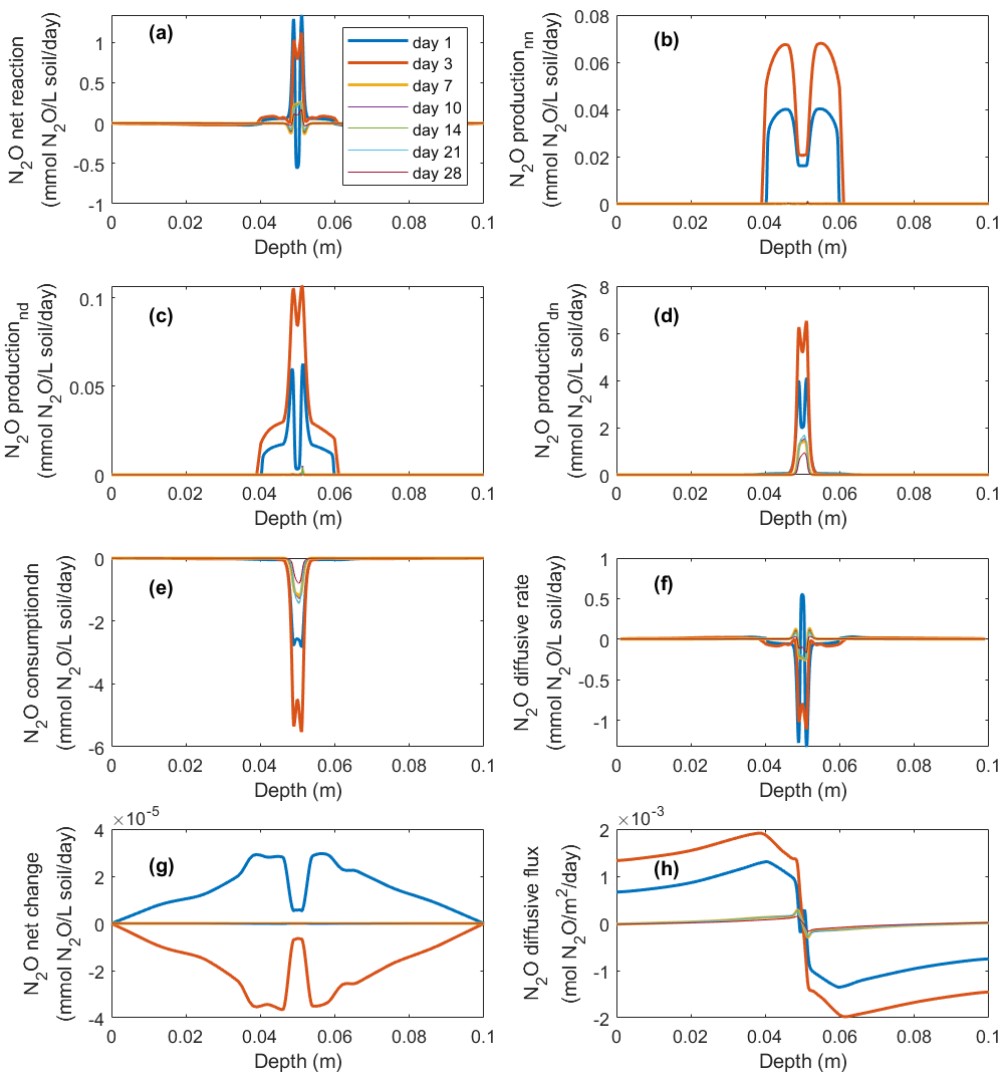

**Figure 5: Simulated N₂O production rate *via* nitrification (nn), nitrifier denitrification (nd), and denitrification (dn), respectively; N₂O reduction rate *via* denitrification (dn); and N₂O change rate *via* diffusion in the -30 hPa treatment. In panel (h), the negative sign represents the downward movement towards the lower soil-air interface (z = 0.1 m), and the positive sign the flow towards the upper soil-air interface (z = 0 m).**

The fate of $NO_3^-$ in the model was determined by three processes: production *via* nitrification, consumption *via* denitrification, and diffusive exchange. According to the model, $NO_3^-$ production mainly occurred outside the saturated zone and peaked around the borders of the manure-saturated zone, where the rate was up to 1.4 mmol $NO_3^-$ $L^{-1}$ soil $d^{-1}$ by day 1, and the peak values continued to increase to reach 2.2 mmol $NO_3^-$ $L^{-1}$ soil $d^{-1}$ by day 3 (Fig. 6a). The increase in $NO_3^-$ production rate continued until day 7 followed by a decrease to the end of the four-week simulation. The consumption of $NO_3^-$ *via* denitrification dominated at the center of the manure zone (Fig. 6b) where the soil was saturated and the $O_2$



availability was low (Figs. S1.4 and S2.5). The $NO_3^-$ consumption rate greatly increased during the first week with a maximum value of around 7 mmol $NO_3^-$ $L^{-1}$ soil $d^{-1}$ on day 3 (Fig. 6b), greater than the simultaneous production rate. A net consumption rate within the manure-saturated zone was therefore simulated (Fig. 6c) based on these two microbial

455    pathways. Besides biochemical reactions, the supply of $NO_3^-$ used for denitrification also came from the process of $NO_3^-$ diffusion, as depicted in Fig. 6d. Driven by the concentration gradient between the manure zone and the bulk soil, nitrification-derived $NO_3^-$ migrated to the center of the manure hotspot where $NO_3^-$ was effectively consumed, as indicated in Fig. 6f. Between day 1 and day 3, the $NO_3^-$ diffusion rate within the saturated zone doubled from ca. 3 mmol $NO_3^-$ $L^{-1}$ soil $d^{-1}$ to the maximum value of ca. 6 mmol $NO_3^-$ $L^{-1}$ soil $d^{-1}$, compensating for the relatively lower nitrification rate to sustain

460    denitrification (Fig. 6d).

In the -100 hPa treatment, the spatial pattern of $NO_3^-$ transformations were different from the -30 hPa treatment, and here high rates of $NO_3^-$ production also occurred at the center of the manure zone (Fig. S3.2) where the water content was relatively higher than in the surrounding soil, but lower than in the -30 hPa treatment. The diffusion rate ranged from 0.8 to 1.6 mmol $NO_3^-$ $L^{-1}$ soil $d^{-1}$ at the center of the manure zone (Fig. S3.2d), and the production rate ranged from 1.8 to 2.8 mmol

465    $NO_3^-$ $L^{-1}$ soil $d^{-1}$ (Fig. S3.2a). Fig. S3.2b shows that $NO_3^-$ reduction during denitrification mainly occurred on day 1 at around 1.8 mmol $NO_3^-$ $L^{-1}$ soil $d^{-1}$, which declined greatly to 0.4 mmol $NO_3^-$ $L^{-1}$ soil $d^{-1}$ during the following two days, and thereafter continued to show a declining trend. A net positive change of $NO_3^-$ concentration was simulated during the first three days, and then a net $NO_3^-$ depletion was predicted in the manure zone after one week (Fig. S3.2e).





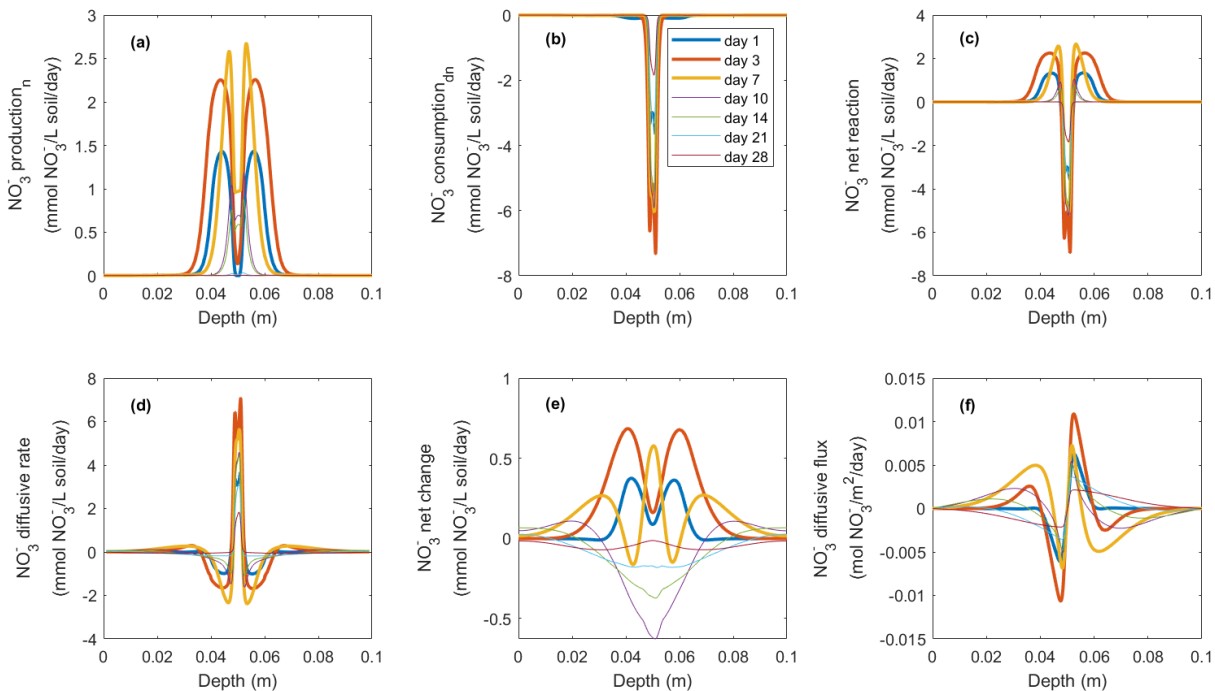

**Figure 6: Individual $NO_3^-$ reaction rates, diffusive rate, net change rate, and diffusive flux in the -30 hPa treatment from model simulation. In panel (f), the negative sign represents the downward movement towards the lower soil-air interface (z = 0.1 m), and the positive sign the flow towards the upper soil-air interface (z = 0 m).**

The biomass of nitrifiers (i.e., AOB and NOB) and denitrifiers increased rapidly during the first week in the simulations (Figs. S2.1-S2.2) and dominated in spatially distinct niches. Thus, growth of nitrifying bacteria mainly occurred inside and around the manure-affected area, with a visible increase in the soil at 6-8 mm from the initial borders of manure application. The biomass of AOB and NOB peaked by day 7 close to the soil-manure interface and were by then, respectively, up to four times as high and twice as high the background level in both treatments. Microbial decay then dominated so that the nitrifier biomass decreased but still remained above the background level by day 28. The denitrifer biomass was initially unevenly distributed within the soil where the 2 mm center of the manure zone had a denitrifier biomass ten times that of the bulk soil to account for denitrifiers existing in the liquid manure, as described in Sect. 2.4.1. The growth of denitrifiers mainly occurred inside the manure zone and peaked at up to 30 times the background level in bulk soil by day 10 in the -30 hPa treatment, which was followed by a temporally stable phase until day 21 and then a declining trend until the end of the experiment.

In the -30 hPa treatment, the $O_2$ consumption rate was well aligned with the $CO_2$ production rate through aerobic respiration (Figs. S3.5b and S3.7b). By day 1 and day 3, $O_2$ consumption by aerobic respiration ($S_{O_2,r}$) showed peak values of -32 and -83 mmol $O_2$ $L^{-1}$ soil $d^{-1}$, respectively, around the soil-manure interface, and simultaneously the $O_2$ consumption *via* nitrification ($S_{O_2,n}$) had peak values of -4.5 and -7.5 mmol $O_2$ $L^{-1}$ soil $d^{-1}$ in the same position. At the center of the manure





zone, $CO_2$ was mainly produced by respiration of denitrifiers during the simulated four-week period (Fig. S3.5c), with up to

490    12 mmol $CO_2$ $L^{-1}$ soil $d^{-1}$ on day 1, in contrast to the aerobic respiration that dominated in most of the core volume. In the -

100 hPa treatment, oxygen consumption was mainly owing to aerobic respiration by day 1 where the peak values of $S_{O_2,r}$

and $S_{O_2,n}$ were -73 and -5 mmol $O_2$ $L^{-1}$ soil $d^{-1}$. On day 3 nitrifier denitrification dominated the use of $O_2$ within the manure-

application zone (peak: -8.6 mmol $O_2$ $L^{-1}$ soil $d^{-1}$) whereas aerobic respiration has greatly declined and was only elevated at

the center (peak: -7.4 mmol $O_2$ $L^{-1}$ soil $d^{-1}$) (Figs. S3.6b and S3.8a). $CO_2$ produced by denitrification peaked at the center of

the manure zone, but was overall less than aerobic respiration during the four weeks in the -100 hPa treatment (Fig. S3.6c).

The oxygen diffusion rates closely followed the consumption rates (Figs. S3.7a, S3.7d, S3.8a, and S3.8d). This indicates that

the overall air porosity within the 10 cm soil core, including a small fraction of air porosity (i.e., 0.01) in the saturated zone,

was able to ensure adequate soil aeration at the oxygen consumption rates occurring in the experiment. The simulated

oxygen content in the pore air began to increase within one day (Figs. S2.1 and S2.2). However, the $O_2$ availability to

enzyme reaction sites in the model depend not only on the oxygen content in soil pore air but also on the soil tortuosity in

relation to the soil water content. As shown in Fig. S2.5, the simulated $O_2$ availability showed dramatic heterogeneity across

the soil profile, with a steep downward slope at the center of the manure zone, but no evident temporal changes were

simulated after day 1 in the two treatments. The -100 hPa treatment in general showed a higher level of $O_2$ availability than

the -30 hPa treatment, but the $O_2$ available in the near-saturated center (i.e., 0.049-0.051 m) was still lower than that in the -

30 hPa bulk soil and thus facilitated local denitrification.

### 3.3 Scenario tests: the importance of solute diffusion in the model

Figure 7 shows the results of four scenarios to investigate the importance of solute diffusion for gas emissions in the baseline

model scenarios and, presumably, measurement results presented above. Switching off the diffusion of all solutes greatly

reduced $N_2O$ and $N_2$ fluxes during the four-week simulation of the -30 hPa treatment were compared to the baseline

simulation and the observations, and the emission peaks during the first three days disappeared (Fig. 7a and 7c). By allowing

the diffusion of $NO_3^-$, but not DOC, $NH_4^+$, and $NO_2^-$ (Scenario 2), $N_2O$ and $N_2$ flux were enhanced to some degree but still

could not explain the temporal patterns in the observation. Allowing $NH_4^+$ diffusion only (Scenario 3) increased the

simulated $N_2O$ and $N_2$ fluxes at the start of simulation compared to Scenario 1, although not as much as the change caused

by allowing $NO_3^-$ diffusion only. The simulated N gas fluxes dropped to negligible levels after one week in Scenario 3. In

Scenario 4, allowing the diffusion of three N solutes, but not DOC, the simulated $N_2O$ and $N_2$ fluxes were greatly increased

compared to Scenarios 1-3, and the temporal pattern, characterized by an early flux peak and a long tail, was similar to the

baseline. However, the timing of peaks came later than in the measurement, and the simulated peak was even higher when

preventing DOC flux between manure layer and soil. Simulated $CO_2$ emission showed a peak around day 2 in the baseline

scenario, which was delayed by approx. one day and declined from ca. $5\times10^5$ to ca. $2\times10^5$ µg C $m^{-2}$ $h^{-1}$ in the four scenarios

(Fig. 7e). Not much difference was found in the simulated $CO_2$ emissions between the four scenarios.





In the -100 hPa treatment, Scenario 1 and Scenario 2 showed similar results in that the early $N_2O$ and $N_2$ fluxes within the first three days were lower than that in the baseline scenario but still higher than the observations (Fig. 7b and 7d). The differences between the baseline and the two scenarios were much smaller than that in the -30 hPa treatment. Whereas the inclusion of $NO_3^-$ diffusion (Scenario 2) did not show any stimulation of $N_2O$ emissions compared to Scenario 1, including

$NH_4^+$ diffusion resulted in a marked increase of simulated gas fluxes at the early stage which was identical with the result of Scenario 4. The measured gas fluxes were overestimated in Scenario 3 and Scenario 4, similar to the tendency in the -30 hPa treatment. The simulated $CO_2$ peak flux was lower than the baseline simulation and was delayed by approx. one day (Fig. 7f).

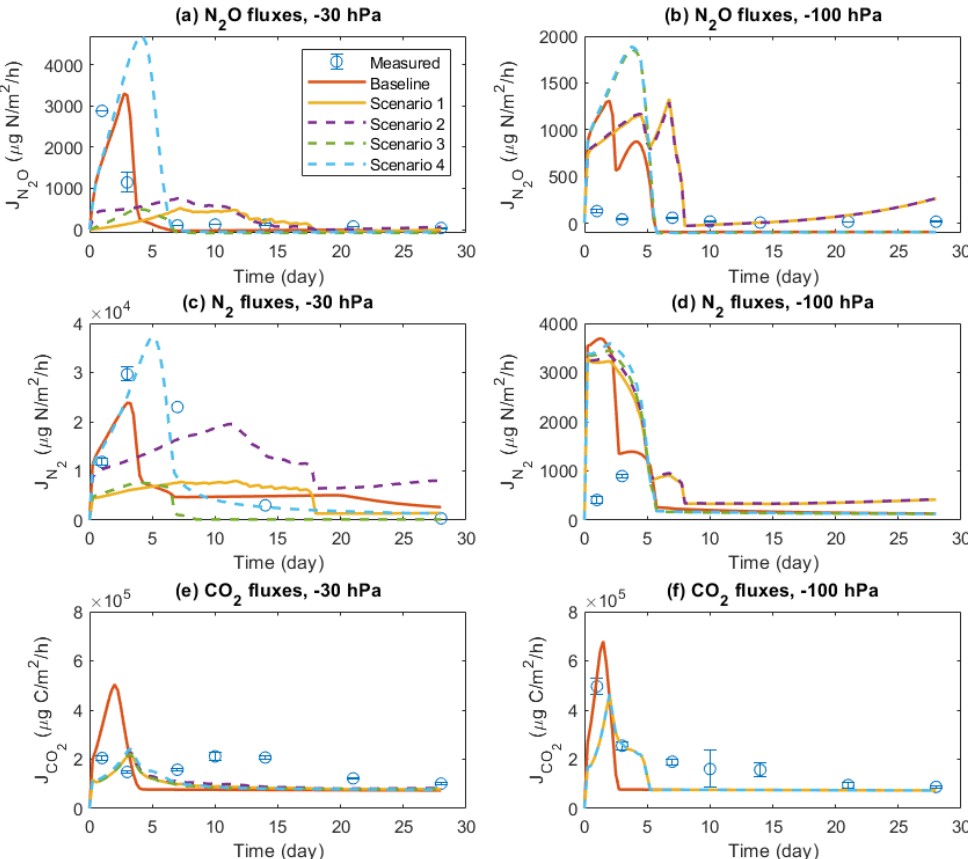

**Figure 7: Comparison of different scenarios with respect to $N_2O$, $N_2$, and $CO_2$ fluxes in the -30 hPa and -100 hPa treatments. In each panel, there are measured data, a baseline simulation where the diffusion of all solutes is included, and four scenarios 1-4, where Scenario 1 does not allow any solute diffusion, Scenario 2 allows only $NO_3^-$ diffusion, Scenario 3 allows only $NH_4^+$ diffusion, and Scenario 4 allows diffusion of $NH_4^+$, $NO_3^-$, and $NO_2^-$, but not DOC.**



## 4. Discussion

**4.1 Model performance**

The model generally reflected the magnitudes of individual $N_2O$ and $N_2$ emissions measured in the experiment where two moisture levels were tested, with the larger model errors in simulating the -100 hPa treatment. Specifically, the model captured well the peak fluxes of $N_2O$ and $N_2$ in the -30 hPa treatment, but overestimated the early stage of gas emissions in the -100 hPa treatment, and temporal dynamics were not always aligned with observations. However, the main purpose of

this study was not to accurately reproduce emissions, but to investigate C and N transformations via major microbial pathways in a soil environment with liquid manure representing a type of organic hotpots that is characteristic of intensive agriculture. The model-data errors under different moisture levels could be related to the use of relationships between soil moisture and diffusivity factors to calculate the availability of substrates at the sites of enzymatic reactions, rather than empirical moisture response functions as process-based models often do. Since the distribution of soil water content in the

soil cores remained almost constant during the four-week incubation (Fig. S1.4), temporal variations of soil moisture responses and gas diffusivity associated with water movement were less important than solute diffusion in the experimental setup used in this study.

The simulated profiles of solutes, including mineral N and DOC, showed generally the same spatiotemporal patterns as the measured or estimated data from the experiment, albeit with some discrepancies. The overestimation of $NH_4^+$ on day 1 (Figs.

3a, 3b, 4a, and 4b) may have several reasons. First, there was some uncertainty about the $NH_4^+$ initial condition set in the model due to a potential for ammonia volatilization from the cattle slurry while infiltrating into the soil during sample preparation. Next, we assumed that a few hours after manure application the $NH_4^+$ content was uniformly distributed within an 8 mm depth interval, but the manure liquid could have spread to a wider range and hence become more diluted. These two mechanisms would lead to overestimation of the initial $NH_4^+$ concentrations in the modeled manure zone compared to the

actual level on day 0 in the experiment. Uncertainty of the $NH_4^+$ adsorption isotherm (Hunt and Adamsen, 1985; Sieczka and Koda, 2016) could be another reason for the high model estimates of $NH_4^+$ in the manure zone if the modeled $NH_4^+$ mobility was lower than the actual mobility in the loamy sandy soil. Ammonium uptake by bacteria as part of osmoregulation could also have contributed to the rapid disappearance of $NH_4^+$ in the experiment, since the osmotic pressure of solutes in livestock slurry can exceed -1000 kPa (Petersen and Andersen, 1996), and hyperosmotic conditions induce bacteria to accumulate

amino acids (Li et al., 2011). Also, immobilization of $NH_4^+$ would occur in connection with the bacterial growth on the basis of DOC (Recous et al., 1990). Processing of $NH_4^+$ by AOB was also included in the model and partly accounted for this process while the related parameter, yield coefficient, was fixed by taking values from the literature where uncertainty existed.

The spatiotemporal pattern of modeled $NO_3^-$ aligned well with measurements in the -100 hPa treatment, while in the -30 hPa

treatment a depletion of $NO_3^-$ was simulated around the manure zone by day 28 which was absent from the $NO_3^-$ profile





observed. The rebound of $NO_3^-$ in the experiment could be related to mineralization of labile organic N in the manure followed by nitrification, a mechanism that is currently not included in the model and a potential source of model structure uncertainty. Møller et al. (2004) reported an average protein content of 150 g kg$^{-1}$ VS in cattle slurry, part of which could have been degraded within the first days of incubation. The modeled denitrifier biomass in the -30 hPa treatment remained much higher (> 23 times) in the manure zone compared to the background level on day 28, whereas the biomass of AOB and NOB had declined to less than twice the background level (Fig. S2.1). In an experimental study by Petersen et al. (1992), the potential for denitrification in the manure-saturated zone decreased in the late stage of the incubation, which was explained by the decay of microorganisms or inhibition of enzymatic activities caused by $NO_2^-$ toxicity, specifically the unionized nitrous acid species (Abeling and Seyfried, 1992). Microbial decay in the model followed first-order kinetics with a constant decay rate and it may be possible in future work to improve relevant parameters and consider the response of substrates on microbial decay and maintenance based on previous studies (Ni et al., 2011).

The spatial stratification of nitrifier and denitrifier growth in the model (Figs. S2.1 and S2.2) was consistent with experimental results (e.g., Nielsen et al., 1996; Petersen et al., 1992) showing a very high potential for nitrification developing within a few millimeters from the manure-soil interface and the potential for denitrification growing within the manure-saturated zone. In both treatments, after significant growth of microbial populations, the reduction in biomass of aerobic heterotrophs, AOB, NOB, and denitrifiers indicated a predominance of decay over growth which had been gradually limited by the depletion of substrates, i.e., DOC for heterotrophs and denitrifiers, ammonium for AOB, and nitrite for NOB, as the simulated nitrogen removal from the soil continued. The spatial distributions of aerobic heterotrophs and nitrifying bacteria were similar and distinctly different from that of denitrifying bacteria.

Variation in the availability of $O_2$ at enzymatic sites was the main factor accounting for the stratification of microbial communities in the model, where the growth of denitrifying bacteria was promoted at the center of the manure zone with $O_2$ limitation while aerobic bacteria preferentially developed outside the saturated or near-saturated zone with better access to $O_2$. The simulated $O_2$ concentration in soil pore air indicated that $O_2$ re-entered the center of the manure zone within the first 24 hours (Figs. S2.1 and S2.2). If the anoxic volume had been larger, as observed in experiments with cm-scale hotspots (Markfoged et al., 2011; Petersen et al., 1996), the $O_2$ depletion zone would probably have been larger and temporally more stable. Thus, micro-sensor measurements showed an oxygen penetration into a manure hotspot of less than 1 mm during the first week (Petersen et al., 1996). The simulated rapid recovery of $O_2$ in the air-filled pores in the manure zone predicted by the model was related to the small extent of the anoxic zone that was predicted for the experimental system, which was as narrow as 2 mm. However, the limited gas diffusivity within the manure zone still led to a low level of $O_2$ available for enzymatic reactions throughout the four-week simulation (Fig. S2.5).





## 4.2 Effects of solute diffusion

Switching off solute diffusion greatly reduced the modeled $N_2O$ emissions, but mainly in the -30 hPa treatment (Fig. 7), indicating that at -100 hPa soil water potential, solute diffusion was not the main limiting factor. The contrasting effects that we found in the scenario tests with complete or partial elimination of solute diffusion were considered to reflect the importance of different mechanisms contributing to $N_2O$ emissions at the two moisture levels.

Dissimilatory $NO_3^-$ reduction by denitrifying bacteria initiates the chain of reactions resulting in a stepwise reduction of nitrogenous oxides to $N_2$, and emission to the ambient air. In the -30 hPa treatment, a hotspot of $NO_3^-$ consumption was present at the center of the manure zone where, by day 1, the diffusive supply was ca. 103 % of the $NO_3^-$ demand by denitrification, and by day 3 ca. 99 % (Fig. S3.9). After day 7, $NO_3^-$ generated from nitrification accounted for a greater proportion of the demand for denitrification and was 12-17 % in the second week. The $NO_3^-$ diffusing from the soil-manure interface towards the anoxic center was due to nitrification outside the hotspot as well as transport of soil $NO_3^-$. Eliminating the diffusion of $NO_3^-$, the initially low $NO_3^-$ at the anoxic center could not be replenished by either source, thereby hindering the potential for denitrification in the simulation. The lack of $NO_3^-$ transport could account for the dramatic decrease in $N_2O$ flux in the Scenario 1 test for the -30 hPa treatment. By day 1, the $NO_3^-$ produced from nitrification, which could be estimated from Figs. 6a and soil moisture, was still lower than the initial $NO_3^-$ content in the bulk soil which was 4.5 mmol $L^{-1}$ in the -30 hPa soil and 6.2 mmol $L^{-1}$ in the -100 hPa soil (Figs. S2.1 and S2.2). Nitrification-derived $NO_3^-$ gradually became comparable to, and even exceeded, the soil $NO_3^-$ level after day 3. However, the transport of $NO_3^-$ from the bulk soil, initially present at z < 0.046 m and z > 0.054 m, to the anoxic zone (0.049-0.051 m) became less efficient than the nitrification-derived $NO_3^-$ which was produced immediately outside the anoxic zone, thus increasing the relative importance of nitrification-derived $NO_3^-$ over time, as shown previously in controlled experiments with [15]N-enrichment of the external $NO_3^-$ supply (Nielsen et al., 1996). Allowing $NO_3^-$ diffusion only (Scenario 2) did not greatly improve the $N_2O$ emission compared to Scenario 1, which reflects the limited role of $NO_3^-$ supply from the bulk soil owing to the increasing transport distance and absence of nitrification without access to $NH_4^+$. However, allowing for $NO_3^-$ diffusion in the -30 hPa soil was still better than allowing $NH_4^+$ diffusion only in this treatment (Scenario 3), and hence the co-occurrence of nitrification and denitrification within the manure zone was insufficient to account for the observed $N_2O$ emissions. These limitations could be alleviated by allowing the diffusion of all nitrogenous substrates from the simulation (Scenario 4), allowing for the coupling between nitrification outside the anoxic zone and denitrification inside this zone.

In the -100 hPa treatment, in contrast, the spatial stratification of nitrification and denitrification was less evident because the production of $NO_3^-$ within the denitrification zone was only moderately lower than the production around the interface and could cover the demand for denitrification in this scenario (Fig. S3.2). Specifically, the nitrification rate and diffusion rate within the 0.049-0.051 m interval were, respectively, 103 and 89 % of the denitrification rate on day 1, and the nitrification rate continued to increase by day 3 (Fig. S3.10), showing that the $NO_3^-$ consumption within the denitrification center in the -



100 hPa treatment was much lower than the $NO_3^-$ supply, with nitrification rather than solute diffusion as the dominating source. Compared with the -30 hPa treatment, the larger air-filled porosity (Fig. S1.3) and higher availability of $O_2$ (Fig. S2.5) in the -100 hPa treatment thus supported a higher nitrification activity which was sufficient to sustain the denitrification activity at the center of the manure zone. The limited solute diffusion in the -100 hPa treatment was due to the much lower effective diffusion coefficients in the aqueous phase, and with more extensive nitrification, gradients of $NO_3^-$ concentrations between the soil and the manure zone could be smaller relative to that in the -30 hPa treatment. The simulated $N_2O$ emission was not as much influenced by cutting off the solute diffusion as that of the -30 hPa treatment, indicating that tightly coupled nitrification and denitrification within the manure zone, rather than solute diffusion from surrounding soil, sustained denitrification. This is in accordance with the results of Manzoni and Katul (2014) who found that hydrological connectivity may continue to exist at microscale even when macroscale solute diffusivity is interrupted. Hence, the model predicted that when manure is applied to drier soil, the redistribution of water may support an even closer coupling between nitrification and denitrification than is the case in wetter soil with much less redistribution of manure liquid (Petersen et al., 2003).

Some experimental studies (Meyer et al., 2002; Nielsen et al., 1996; Petersen et al., 1991, 1992) have reported that in manure-amended soil there may be a distance of only a few millimeters between active nitrification and denitrification zones, facilitating coupled nitrification-denitrification. These observations are consistent with our model predictions for the -30 hPa treatment where a very pronounced stratification of microbial populations was simulated. We also found support for an even closer, though short-lived, coupling of nitrification and denitrification in the simulation of the -100 hPa treatment. The manure hotspots investigated here were relatively small with only a 2 mm layer dominated by manure, and field application of liquid cattle manure is likely to form, depending on application method, cm-scale hotspots with a greater potential for developing coupled nitrification-denitrification. Based on the discussion above, we suggest that solute diffusion should be considered a key control of denitrification and $N_2O$ emissions in soil-manure systems. The extent will depend on factors such as soil water content and texture controlling water film continuity, and manure application rate and application method determining soil-manure contact and stability of anoxic volumes that can support denitrification.

### 4.3 Model uncertainties and future applications

The model presented was able to qualitatively reflect gaseous emissions and changes in carbon and nitrogen pools in and around an organic hotspot in soil, but several aspects of model uncertainty remain. First, the initial condition used in the model was a simplification of actual conditions due to lack of detailed soil analysis within the first 24 hours of incubation. We considered the stabilized slurry redistribution as a "box", i.e. manure liquid was redistributed by piston flow away from the initial placement of the added manure. However, in reality the joint effects of dispersion and diffusion could result in a more continuous distribution of dissolved slurry components over a wider zone. Also, some reactive C and N components contained in the manure may have been lost between the time of manure sampling for chemical analyses and the initiation of





incubations. These two factors could result in the estimated initial concentrations of manure components in the assumed
manure zone being overestimated. We suggest that if incubation experiments are designed to support modeling, more
frequent sampling early in the experiment can provide an opportunity to better characterize the behavior of the system.

In the model, we assumed the gases generated from biochemical reactions existed in air-filled pores instead of modeling the
water-air exchange. Therefore, the current model is suitable for simulating processes in unsaturated rather than saturated soil
where $N_2O$ and $CO_2$ can only exist in dissolved form. However, when saturated volumes only account for a small fraction of
the total samples, small adaptations can be made in model application. In this study, we included a small fraction of air-filled
porosity (i.e. 0.01) in the 2 mm saturated zone when modeling the -30 hPa treatment to ensure the validity of Eq. 1 over the
entire model domain. The value of the air-filled porosity did influence the absolute values of simulated gas fluxes. However,
when the value of the air-filled porosity was kept within a reasonable range, specifically, well below the air-filled porosity
(i.e., 0.042) at the center of the -100 hPa treatment samples, the dominating processes associated with the emissions were not
much influenced. Thus, when keeping the zero air-filled porosity in the saturated zone or changing the added air porosity to
be 0.001, the modeled $N_2O$ peak, compared to the baseline simulation and observations, showed an increase in both cases in
the -30 hPa treatment, as anoxia dominated more at the center of soil cores, whereas the qualitative results regarding solute
diffusion and the spatiotemporal patterns of reaction rates remained the same (Figs. S5.1 and S5.2). Based on the above two
air-filled porosity settings in the -30 hPa treatment, we also attempted to re-optimize the model parameters in order to
specifically reduce the model-data errors, and the main conclusions we obtained were not changed (data not shown).

The model was able to depict the spatiotemporal patterns of the main $N_2O$ forming processes in soil including nitrification,
nitrifier denitrification, and denitrification. However, uncertainty exist in the relative contributions of these processes,
especially the nitrifier denitrification pathway, as relevant parameters were not well constrained by the available
measurements, and the simulated gas fluxes were less sensitive to the parameters regarding nitrifier denitrification than to the
parameters regarding the other two processes (data not shown). Detailed quantification of $N_2O$ sources from different
processes was beyond the scope of our study, but isotopic signatures of $N_2O$ is a promising tool in this respect, as shown in
the modeling study by Chang et al. (2022). Besides, when optimizing parameters, manually adjusting the weights in the
objective function may have led to biased model-data errors in some variable estimates (Konak et al., 2006). Exploring better
designs for multi-objective optimization (e.g., Cheng et al., 2002; Konak et al., 2006; Li et al., 2010; Nguyen et al., 2019),
which involve not only temporal variation of gas emissions (e.g., $CO_2$, $N_2O$, and $N_2$) but also spatiotemporal variation of
substrates (e.g., DOC and inorganic N) in the soil, is a topic for further studies to improve $N_2O$ model performance.

The contributions of different biochemical pathways to $N_2O$ emissions have been elaborately studied in wastewater systems
(e.g., Chen et al., 2019; Ni et al., 2011; Peng et al., 2014); and the role of solute diffusion has been a long-time topic in the
experimental and modeling field of soil aggregates (e.g., Kremen et al., 2005; Schlüter et al., 2018; Smith, 1980) and
permeable sediments (e.g., Kessler et al., 2013). These processes, however, have not been well explored in meso- and large-





scale soil modeling. Future applications of the model could include exploring how different spatial distribution patterns of manure hotspots could influence the prediction of $N_2O$ emissions from soil. Also, the influence of soil moisture associated with modeled diffusional constraints on soil $N_2O$ emissions could be re-assessed based on properly designed experiments.

**5. Conclusion**

Understanding the spatiotemporal distribution of nitrification and denitrification in agroecosystems, and particularly that associated with field-applied manure, may be crucial for reducing the uncertainty of agricultural $N_2O$ emission estimates in process-based soil biogeochemistry models. The present modeling study, supported by an incubation experiment, gives new insights into the importance of biochemical processes in heterogeneous soil environments. The simulation results confirmed

previous experimental work suggesting that nitrifier and denitrifier communities develop in closely coupled, but separate niches in soils amended with manure. Denitrifiers mainly developed in the predominantly anoxic zone whereas nitrifiers and aerobic heterotrophs proliferated around the interface between bulk soil and manure-amended soil. The stratification was affected by soil moisture and became tighter with decreasing soil water content in response to better aeration and constraints on solute diffusion. In accordance with the spatial features of bacterial communities, the saturated or near-saturated center of

the manure zone was a hotspot of $N_2O$ production by denitrification and possibly nitrifier denitrification, whereas nitrification dominated the $N_2O$ production at the better-aerated interface according to the model. In the manure-amended soil volume where $NO_3^-$ was initially depleted, the $NO_3^-$ demand of denitrification was largely maintained by nitrification, demonstrating a strong coupling of nitrification and denitrification either in the same layer or separated by a short distance depending on soil water content. Breaking down the sources and sinks of $NO_3^-$ in the model clearly revealed the important

contribution of solute diffusion to the supply of $NO_3^-$ for denitrification, and neglecting this process will significantly deteriorate the accuracy of $N_2O$ emission estimates, as demonstrated with scenarios without the concurrent transport and reactions of $NO_2^-$, $NO_3^-$, and $NH_4^+$ in the vicinity of organic hotspots. The implications of this study are of importance for soil $N_2O$ modeling in general by suggesting that if solute transport is not included in process-based models when simulating stagnant soil conditions, and organic hotspots are present, then model-estimated $N_2O$ emissions could be much lower than

the actual emissions.

**Code availability**

The codes of the model are available from the authors on request.

**Data availability**

The laboratory experiment and the simulation data are available from the authors on request.



**Author contributions**

JZ and SOP conceived the study. JZ developed the model and carried out the modeling experiments, with contributions from SOP and WZ. ELK conducted the lab experiment. JZ prepared the manuscript and figures with contributions from all co-authors.

**Competing interests**

The authors declare that they have no conflict of interest.

**Acknowledgements**

The authors wish to thank the Danmarks Frie Forskningsfond (grant no. 0136-00118B) for the financial support for this study, and thank Professor Per-Erik Jansson for the insightful feedback. WZ was supported by grants from the Swedish Research Council VR (2020–05338) and Swedish National Space Agency (209/19).

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
