# Peer review of "Modeling coupled nitrification-denitrification in soil with an organic hotspot"

_Biogeosciences, 2023_

## Author Response (AR1)

**Reply on RC1**

Jie Zhang et al.

We would like to thank Prof. E.A. Davidson for his interest in our study, and for the feedback provided. We appreciate these constructive and specific comments, which will help improve the quality of the manuscript. Please find the response to each comment below.

**General comment:**

Zhang et al. (2023) tested a model of coupled nitrification-denitrification with results from an incubation experiment of soil cores in which manure-dominated zone was layered within the cores. The experiment was conducted at two different moisture contents: -30 and -100 hPa. In addition to comparing the model simulations to measurements of N2O, N2, CO2, ammonium, nitrate, and nitrite, the authors also analyzed four simulation scenarios in which diffusion of DOC, ammonium, nitrate, and nitrite were turned on or off in the model run. The manuscript is well written and the results are novel. The authors state that the purpose is not so much developing a model that fits the data, but rather "but to investigate C and N transformations via major microbial pathways in a soil environment with liquid manure representing a type of organic hotpots that is characteristic of intensive agriculture." Through their scenario analysis, they demonstrate that at the higher moisture content, the manure-dominated layer becomes anaerobic and is the site of denitrifier production of N2O and N2, which is dependent upon diffusion of ammonium to nitrifiers in the overlying oxic zone and diffusion of nitrate to denitrifiers from the oxic zone to the anaerobic manure layer. At lower soil moisture, in contrast, the manure-dominated zone receives sufficient O2 diffusion to it so that nitrification and denitrification can be coupled within the same layer.

The model that combines microbial kinetics with diffusive flux of O2 and solutes (Eqs. 5, 23, 24) is based on the concepts of the Dual Arrhenius Michaelis-Menten (DAMM) model of Davidson et al. (2012). Similarly, Sihi et al. (2020) also applied the DAMM model to CO2, N2O, N2, and CH4 emissions and validated the model with field flux data. Equations 12, 19, and 21 of the present manuscript are nearly identical to those described in the supplemental information file for Sihi et al. (2020), except that the present manuscript also includes equations for changes in the denitrifier microbial biomass. Another difference is that Sihi et al. (2020) used probability density functions (PDFs) to evaluate soil microsite heterogeneity within a single soil layer, whereas the present study used cores packed to have vertical variation among layers. Sihi et al. (2020) used a Bayesian approach to parameterize the model, constrained simultaneously by CO2, CH4, and N2O flux measurements, whereas the present study assigned several fixed parameters and fitted others through optimalization of partial differential equations (PDEs). Both modeling approaches show promise for improving our understanding of hot spots and hot moments of trace gas emissions from soils by combining simulations of microbial kinetics with diffusive transport of solutes and gases.

**Response:** We are grateful for these positive comments and the informative summary of the background with reference to related modeling work. As you mentioned, the current study does follow the concepts of the Dual Arrhenius Michaelis-Menten (DAMM) model proposed by Davidson et al. (2012), which uses a series of linked equations to combine the effects of soil water content and soluble substrate supply on carbon and nitrogen transformations in a unifying framework. The description of reaction velocities in the manuscript has been inspired by, or adapted from, the work of Sihi et al. (2020), Chang et al. (2022), and Chen et al. (2019) to suit our special interest in the detailed N transformations and microbial population dynamics. In contrast to the field-scale application by Sihi et al. (2020), the current study is a mesoscale application of the parsimonious DAMM model framework, combining biochemical reactions with diffusive transport of components and aiming to improve our understanding of explicit organic hotspots and factors controlling gaseous emissions.

**Specific comments:**

My suggestions for improvements of the present manuscript include the following:

**Comment (1):** For the baseline model, a sensitivity analysis of the parameterization (both assigned and fitted parameters) would be helpful and probably insightful. For example, overestimation of N2O and N2 fluxes at -100 hPa could be due to inadequate sensitivity of O2 inhibition to denitrification (kI terms) or to underestimation of O2 diffusion (parameters related to gaseous diffusivity and soil porosity). The latter may also affect CO2

**Response:** Thank you for the advice. The parameters chosen for model calibration were determined using an informal sensitivity analysis. However, the details of sensitivity analysis were not mentioned in the present manuscript. In the revised version, we conducted a formal local sensitivity analysis based on the baseline model to examine the impact of the assigned and fitted parameters on the total emissions of each greenhouse gas. The results were added in the supplement Figure S7.1 and shortly discussed in LN 710-713.

**Comment (2):** For relatively complex models such as this one, an analysis of collinearity for the fitted parameters would be helpful to assess equifinality.

**Response:** We agree with the reviewer's recommendation to test the fitted parameters' equifinality. In the revised version, we did an analysis of equifinality based on the posterior parameter ensemble because the best 1% runs (33 runs) exhibited almost no variation in relative RMSE. To analyze the extent of equifinality for the final optimization, we showed the correlation matrix in the supplement Figure S7.2 and shortly discussed in LN 716-721.

**Comment (3):** Would it be possible to fit the data for CO2, N2O, and N2 fluxes simultaneously, rather than sequentially fit for CO2 and then for the nitrogen gases and solutes? You could be getting the right fit for CO2 for the wrong reasons (equifinality), which would then affect the nitrogen simulations in a way that could obscure insights into the N cycling processes. Combining into a single fitting step adds additional constraints that may help minimize spurious parameter fitting.

**Response:** Yes, it is possible to fit the data for the three gas fluxes and solutes simultaneously, and we did that in preliminary tests. The final parameter fitting was done sequentially – firstly for CO2 and then for other gases and solutes together – to accelerate the convergence in the optimization procedure by reducing the number of fitted parameters, but also to improve the overall model performance of the N module. Thus, when we tried to fit the four gas fluxes first and omitted the solutes, all gases fit well but the simulated solute concentrations did not match the data. The sequence used was decided by considering that the connection between soil respiration and N-related processes is not as strong as the inter-connections between N-related processes, as shown in Fig. 1c, and interactions between the three C-related parameters and N-related parameters were considered to be low. The C-related parameters fitted in the final version were similar to the fitted values obtained in the preliminary tests that included N gases, and therefore we consider that the fit for CO2 was robust.

**Comment (4):** The wording on lines 597-598 is a bit misleading. While it is true that, at -100 hPa, solute diffusion was not limiting for N2O production within the manure layer, it was limiting between layers. In contrast, at -30 hPa, solute transport was not limiting between layers, but O2 transport was limiting, requiring that nitrate produced in the overlying oxic layer had to diffuse to the anaerobic manure layer for coupling to occur. Hence, where the nitrification-denitrification coupling occurs (within a layer or between layers) is dependent on both solute and gaseous diffusion.

**Response:** Thank you for the insightful comment. We agree with the reviewer regarding the different roles of solute diffusion at the two water potentials and the importance of gaseous diffusion. We revised the sentences in LN 616-618 to clarify: "Switching off solute diffusion between layers greatly reduced the modeled $N_2O$ emissions in the -30 hPa treatment (Fig. 7), whereas at -100 hPa soil water potential the effect was much less and indicated that solute diffusion between manure and soil layers was already low."

**Reply on RC2**

Jie Zhang et al.

Great thanks to Reviewer 2 for the time and efforts you took to provide useful feedback for our manuscript. The comments and suggestions provide valuable input for revising and improving the paper, and our responses are outlined below.

**General comment:**

The manuscript "Modeling coupled nitrification-denitrification in soil with an organic hotspot" is a model study to better understand and predict N2O after manure application. The overall manuscript is in a good shape and an important contribution to the field. I recommend the publication after some minor revision.
The study aimed to investigate the importance of solute diffusion for the production of N2O in organic hotspots. This is done by a mulit-species, reactive transport model to predict N2O+N2 emissions from an incubation experiment with hot-spots induced by manure and two different moisture conditions.
The model was able to predict N2O+N2 emissions reasonable well, to fit a perfect line to the measured emissions was not the first goal of the authors. By considering all relevant biochemical processes through a set of partial differential equations the authors showed, that there is the need to consider solute diffusion of several species (DOC, nitrate, ammonium and nitrite) to reasonable predict N2O emsissions.

**Response:** Thank you for the positive evaluation and accurate understanding of our work.

**Specific comments:**

However, to assure reproducibility, while the model description is detailed and has all relevant information, there are some more details needed for the experimental design (see below). In addition, here are some specific comments:

**Comment (1):** Add a sentence about the "scenario tests" at the end of the introduction

**Response:** Thank you for pointing out this missing aspect in the introduction. In the revised version, we mentioned the scenario tests in LN 112.

**Comment (2):** L122: In that sentence they were not subscripts and C is not defined for equation (1)

**Response:** Thanks for pointing this out. In the revised version, we replaced the word "subscripts" by "letters" and will note that $C_\gamma$ means the concentration of substrate γ.

**Comment (3):** L43: "hotspots" instead of "hotspot areas"

**Response:** OK, we changed "hotspot areas" to be "hotspots".

**Comment (4):** L93: "This" relates to two sentences above – maybe extend the sentence

**Response:** Sorry for the confusion. We extended the sentence to read: "This common model design for solute transport can be expected to explain…"

**Comment (5):** Your model assumes no diffusion of microbes. Enzymes can be transported through water (e.g. DOI: 10.1016/j.soilbio.2022.108633). Out of curiosity, would this change your model outcome drastically?

**Response:** It is true that our model assumes no diffusion of microbes but rather microbe and enzyme attachment to soil particles. Microbial cells and enzymes have low mobility in partly drained soil due to their high affinity to fine particles and organic surfaces. Although enzymes can be transported in water, the current model is a reaction-diffusion model and does not include convective transport, which suits the experiment we investigated. As water flux was negligible during the experimental incubation, its effects on enzyme transport could be ignored. It is possible to include enzyme diffusivity in the model, but this requires a proper understanding based on experimental proof, which has no consensus yet. Free enzymes in soil pore water are more prone to being denatured or consumed than enzymes attached to surfaces, and adding a limited mobility of microbes and enzymes in the model is not expected to dramatically change the predicted microbial populations and substrate concentrations.

**Comment (6):** Eq: 20, 21,33 34: Delete the l in O2l

**Response:** Thanks for pointing out the typo and we have deleted the l in O2l in relevant equations.

**Comment (7):** S6.1 is not referred to in the main text

**Response:** Thanks for pointing this out. We referred S6.1 in LN 258: "See the supporting information (Sect. S6.1 and Sect. S6.3) for details."

**Comment (8):** Fig. 1. Explain (-) and (+) notation in the legend, maybe add the height of the soil core to the figure

**Response:** The notation (-) and (+) in Fig. 1 indicate the negative or positive response of the process to O2 or DOC and we have added that in the caption. We also noted the height of the soil core to the figure.

Experiment:

**Comment (9):** L205: How did you pack the cores in detail to assure no layering. Where the two stacked cores sealed to prevent O2 diffusing from the side?

**Response:** We prepared the cores by stepwise packing of 1.25 cm layers of soil to a height of 5 cm, at each step adding water and scratching the soil surface to increase the contact between layers. Two cores in identical cylinders were stacked and sealed with adhesive tape, while the top and bottom surfaces were covered with perforated parafilms and thus connected to the air. These details will be presented in a separate publication.

**Comment (10):** L206: Collected from the topsoil?

**Response:** The soil was collected from the plough layer (0-25 cm).

**Comment (11):** L218 More information is needed about the 15N labeling (e.g. how is it applied, how much etc).

**Response:** Two different solutions of 15N-labelled KNO3 were prepared and added during packing, as explained above, to ensure the same final concentration at both water potentials. This information has been added in LN 219-221 in the revised version.

**Comment (12):** How many replicates did you have, how many separate cores did you use for the determination of nitrate etc. during the experiment.

**Response:** For gas sampling, we had three replicates and for soil sampling to determine nitrate etc., we used two cores as replicates. We have added the information in LN 214-219.

**Comment (13):** L243: A bit confusing to use "m" here and above "cm". You could also write something along: " The soil core in the model had the same size than in the experiment with the manure in it's center"

**Response:** OK, we have revised the sentence to avoid confusion with respect to the unit by using "cm" here and the suggested sentence. In LN 245-247, the revised sentence is read: "The soil core in the model had a depth of 10 cm from the top (z = 0 cm) to the bottom surface (z = 10 cm), and the center of manure application was at 5 cm. The soil core in the model had the same length as in the experiment and with the manure at its center."

**Comment (14):** L327-329: Please rephrase

**Response:** OK, we rephrased the sentence as: "Since the peak N2O flux in the -30 hPa treatment was of particular interest, we included also a term representing peak flux error, $(peak_{obs} - peak_{sim})/peak_{obs}$, in the objective function to ensure this interest was met. Here, $peak_{obs}$ and $peak_{sim}$ indicate the maximum daily N2O flux found in the experiment and the model, respectively."

**Comment (15):** L343: What does "a certain degree" mean

**Response:** "a certain degree" means that the degree to which solute diffusion was eliminated depends on the scenario setup as shown in Table 1. We revised "a certain degree" to "different degrees".

**Comment (16):** Figures in general: What do the errorbars represent? SE or SD?- Add this information to the legends as well as the number of replicates

**Response:** The error bars represent standard deviation (SD). We have added this information as well as the number of replicates to the caption.

**Comment (17):** L420: "that" the rate of N2O concentration change "is "caused by gas diffusion

**Response:** Yes, we took the advice and revised the sentence to be: "… indicate that N2O concentration change is caused…".

**Comment (18):** L497: Maybe air-filled porosity or air-filled pore space instead of air porosity

**Response:** Thanks for pointing out. We used "air-filled porosity" instead of "air porosity" in the revised version.

**Comment (19):** L646: It would be good to extend the discussion here with 2-3 sentences to highlight which situation the model and the respective experiment covers best or where do we potentially need follow up (field) experiments and modelling: E.g. at -30hPa on the field it is likely that there is convective transport / how comparable are the structure on the field with the repacked soil – change in tortuosity, connectivity etc. / to which situation / management (e.g. incorporation of manure by tillage, injection) is this experiment closest

**Response:** Thanks for the advice and it is a good idea to briefly discuss the aspects you mention. The experiment represented a period after the incorporation of liquid manure with no rainfall causing infiltration or leaching around the surfaces. In accordance with this, the model simulated soil conditions with constant soil moisture levels below or at the water holding capacity (i.e., no leaching). Water convection, e.g., during and after rainfall or irrigation, will require an extension of the model concept by adding hydrological processes. "Hotspot" effects depend not only on application rate, but also on the application method defining the contact area between soil and manure. The model may be used to predict effects of surface-to-volume ratios of manure-amended soil corresponding to different application methods (e.g. incorporated by ploughing, or injected), which may then be examined in experiments under field conditions. In the revised version, we have modified the discussion to highlight these points in LN 732-747.

**Reply on RC3**

Jie Zhang et al.

We would like to thank Reviewer 3 for the valuable feedback for our manuscript. We really appreciate these knowledgeable and constructive comments, which will be very helpful for revising and improving the paper. Please find the response to each comment below.

**General comment:**

MS "Modeling coupled nitrification-denitrification in soil with an organic hotspot" by Jie Zhang et al. presents a novel modeling approach validated against experimental data obtained in a specially designed experiment simulating a denitrification hotspot in the form of a manure patch. The paper contributes to our understanding of the drivers that determine the high spatial and temporal variability of N2O emissions and other relevant processes in the case of agricultural soils amended with manure. Main conclusions of the authors derived from the model exercise and scenario runs point out that not only gaseous diffusion, but also distribution and diffusion of liquid substrates (NO3-, DOC) have to be considered for proper modeling of the processes. The article brings new scientific information, it complements the previous publications on the topic.

The complicated model structure requires justification of new pools added and should be compared with existing model approaches. Explicit modeling of the biomass of different microbial subgroups such as AOB and NOB in case of nitrification seems excessive to me, since the real biomass and activity of these groups will hardly be verified by experimental measurements. Even if the gene abundances characterizing these microbial groups are known, the expression of the corresponding enzymes and their contribution to nitrification remain unclear. In addition, there are microorganisms (nitrifiers) capable of performing multiple steps, and most soil denitrifiers are facultative anaerobes that can rapidly switch from respiration to denitrification under O2 limitation. This means that for model parsimony, the number of biomass pools could be minimized, and I think without much effect on model outputs. It would be relevant to compare your model approach with existing ones where nitrifying microorganisms were not split, e.g. Blagodatsky et al., 2011.

Section 3.1. The results of the model fit for gaseous emissions look a bit disappointing, since one of the tasks of the model is to properly predict GHG emissions, and this type of prediction is mediocre. I would suggest a quantitative estimation of model performance and/or model uncertainty using either very simple statistical estimates like RMSE and Nash-Sutcliffe model efficiency or more advanced methods like Bayesian parameter space estimation. My opinion, based on a general feeling after reading your MS (without quantitative estimates), is that your model is over-parameterized. The comparison of scenario runs could also be done on a quantitative basis using the mentioned approaches and AIC estimation.

**Response:** We thank the reviewer for the knowledgeable and constructive comments. We agree that explicit modeling of different microbial groups such as AOB, NOB is not commonly found in soil N2O models. However, we

consider that developing soil N2O models that include explicit microbial populations and individual processes of nitrifier nitrification, nitrifier denitrification, and denitrification should be a good future practice that will allow modelers to incorporate new knowledge about microbial transformations into models. In the field of wastewater treatment, which is also an important source of N2O emissions, mathematical N2O modeling often includes the explicit representation of different groups of nitrifiers and denitrifiers, which has reached a maturity that allows for proper estimation of site-specific N2O emissions and developing mitigation strategies (Ni and Yuan, 2015). In a recent study, Chang et al. (2022) developed a detailed mechanistic soil N2O model that includes explicit microbial populations of nitrifiers and denitrifiers. Such advances could potentially improve the precision of modeling N2O emissions under various environmental conditions.

In model calibration, we optimized the parameters by reducing the sum of relative RMSEs of multiple variables but did not show that statistics in the results. In the revised version, we have added statistical estimates for the gas emission results to indicate the model performance in the supplement Table S8.1. However, as stated in the manuscript, the prediction of N2O emissions was only one of several optimization criteria considered.

Specific comments:

**Comment (1):** L265-267: Both assumptions are vague, as N mineralization always occurs during microbial die off (included in the model) and immobilization by AOB and NOB is relatively low compared to denitrifying and aerobically growing microbial biomass.

**Response:** Thanks for the comment. We did not include the mineralization of organic N from POC in the model as previous studies have associated high C/N with reduced N mineralization in slurry. On the other hand, we agree that the microbial fraction of POC and SOC has a lower C/N ratio where N mineralization occurs during microbial die-off, and including this pool is a topic for future research. However, in the manure hotspot environment the N available for nitrifiers is dominated by ammonium derived from urea in excretal returns or N mineralized during storage.  In the simplified description of microbial growth, which follows the the study by Chen et al. (2019), AOB and NOB growth was linked to N incorporation while heterotrophic growth was linked to C incorporation. We extended the discussion in LN 700-708 to clarify these points.

**Comment (2):** Section 2.4.5 (lines 341-346). Analysis of diffusion fluxes of NH4+ seems to me a bit strange, as you already included retardation factor for NH4+ (Eq.3). The simultaneous application of two different ways for NH4+ distribution looks excessive. Instead of switching off NH4+ diffusion you could adjust the adsorption factor in Eq.3.

**Response:** We consider that adsorption and diffusion of non-sorbed NH4+ are two different processes, and the analysis of diffusive NH4+ flux is not in conflict with its retardation property. Also, if we had adjusted the adsorption factor instead of switching off NH4+ diffusion, it would not be comparable to the scenario tests with other N species.

**Comment (3):** L537: The model errors were not quantified, and this could be done to support the statements here.

**Response:** We added statistical estimates for model errors to support the statements in Table S8.1 and in LN 550-559.

**Comment (4):** L539-541: Not all major C and N transformations were estimated in the experiment and therefore authors cannot judge the model success against these pools. Microbial C and N dynamics would be helpful in this respect, but they were not measured and cannot help in verification of major C and N fluxes. As far as I can see, immobilization of NH4 and NO3 in microbial biomass was not fully presented in the model (e.g. Eq. 45-48), but these processes could have definitely higher rates than nitrification, for example.

**Response:** The focus of the study was on the coupled nitrification-denitrification that induces the spatiotemporal dynamics of multiple C and N species, and the consideration of additional microbial processes, e.g., immobilization, would further complicate the system. However, based on the current work it is possible to extend the model structure by adding the immobilization of NH4+ and NO3- into microbial biomass if its importance to the investigated system is justified by experimental work. Earlier studies have shown that it is possible to describe microbial C and N dynamics, and exponential growth of nitrification and denitrification potentials, around soil-manure interfaces at mm-scale resolution (Frostegård et al., 1997; Petersen et al., 1992), and new studies with this focus could support model development. We extended the discussion in LN 700-708 to clarify these points.

**Comment (5):** Discussion: lines 648-651: The conclusion concerning the importance of diffusion processes consideration in modeling of denitrification and N2O emission is fully grounded, when manure - soil interface is of interest. And the current MS contributes well in solving of this research task.

**Response:** Thank you for the positive comment on our work.

Technical comments:

**Comment (6):** L15: ...different microbial populations...

**Response:** OK, we have revised the sentence.

**Comment (7):** L121-128: It is not stated in the description what is z and t. I presume that these are vertical dimension and time, but this need to be stated. Do you consider vertical gradient dependent diffusion? As experimental design has not been described yet, it is not clear.

**Response:** Thank you for pointing this out. Yes, z and t are soil depth and time and we have added that in the revised version. We consider the vertical dependent diffusion along the soil depth and the model is a one dimensional model.

**Comment (8):** L170-174: The inhibition effect of substrates is not included in the model formulation; therefore, this description should be moved to discussion. At this place it detracts from the understanding of the main model equations.

**Response:** OK, we have moved this part to the discussion in LN 698-700.

**Comment (9):** L190-193: It is not clear, which part of biomass then will be part of equations determining reaction rates (e.g. 12-15 and 19-20)?

**Response:** The sum of the two parts of microbes is part of equations and in the revised version, we used $B_{base}$ and $B_{new}$ to indicate the two parts in the revised version. The total biomass $B = B_{base}+B_{new}$ is used in Eqs. 12-15 and in Eqs. 19-20. The newly produced biomass $B_{new}$ is calculated in Eqs. 25-28.

**Comment (10):** L241-242: It would be better to define precisely what means deficit in terms of oxygen concentration units.

**Response:** The oxygen deficit means zero oxygen concentration and we defined it more precisely in LN 243-244.

**Comment (11):** L250-255: It is unfortunate that DOC was not measured in the beginning of experiment. It is relatively simple and would help in model initialization.

**Response:** We agree, but unfortunately DOC was not quantified, and instead we used a conversion factor between SOC and DOC for the estimation, a strategy inspired by previous studies (Davidson et al., 2012). However, DOC measurement is encouraged for future applications of the model.

**Comment (12):** L257-259: Similar to previous comment I should say that DOC and TOC content of applied manure would be needed for reliable evaluation of the model. Especially considering the highly variable DOC in manure, depending on composting time and initial properties.

**Response:** We agree that measurements of DOC and TOC of applied manure is helpful for model evaluations to reduce input uncertainty. This will also be important for modeling of manure treatment such as anaerobic (co)digestion or separation, where accurate model description of treatment effects on organic matter pools will be essential.

**Comment (13):** L274: See my previous comment concerning O2 limitation - precise value is needed.

**Response:** OK, we have revised the sentence in LN 278 to precisely define the O2 deficit which is zero concentration.

**Comment (14):** Figure 5: It would be good to present the processes responsible for the N2O emission directly at the figure, not just as an abbreviation at y-axes.

**Response:** OK, we have revised the figure and presented the relevant processes directly in Figure 5.

**Comment (15):** Figs in supplement: in many figures in supplementary material there is no full legend and only letters for subplots are present. You should either add full legend in figure capture or, better, as was mentioned ina my other comment make clear label of described process directly on subplots.

**Response:** OK, we have added full panel captions in those figures in the supplement section S3.

**References**

Blagodatsky, S., Grote, R., Kiese, R., Werner, C. and Butterbach-Bahl, K.: Modelling of microbial carbon and nitrogen turnover in soil with special emphasis on N-trace gases emission, Plant Soil, 346(1–2), 297–330, doi:10.1007/s11104-011-0821-z, 2011.

Chang, B., Yan, Z., Ju, X., Song, X., Li, Y., Li, S., Fu, P. and Zhu-Barker, X.: Quantifying biological processes producing nitrous oxide in soil using a mechanistic model, Biogeochemistry, 159, doi:10.1007/s10533-022-00912-0, 2022.

Chen, X., Ni, B. J. and Sin, G.: Nitrous oxide production in autotrophic nitrogen removal granular sludge: a modeling study, Biotechnol. Bioeng., 116, 1280–1291, doi:10.1002/bit.26937, 2019.

Davidson, E. A., Samanta, S., Caramori, S. S. and Savage, K.: The Dual Arrhenius and Michaelis – Menten kinetics model for decomposition of soil organic matter at hourly to seasonal time scales, Glob. Chang. Biol., 18, 371–384, doi:10.1111/j.1365-2486.2011.02546.x, 2012.

Frostegård, A., Petersen, S. O., Bååth, E. and Nielsen, T. H.: Dynamics of a microbial community associated with manure hot spots as revealed by phospholipid fatty acid analyses, Appl. Environ. Microbiol., 63(6), 2224–2231, doi:10.1128/aem.63.6.2224-2231.1997, 1997.

Ni, B. J. and Yuan, Z.: Recent advances in mathematical modeling of nitrous oxides emissions from wastewater treatment processes, Water Res., 87, 336–346, doi:10.1016/J.WATRES.2015.09.049, 2015.

Petersen, S. O., Nielsen, A. L., Haarder, K. and Henriksen, K.: Factors controlling nitrification and denitrification: a laboratory study with gel-stabilized liquid cattle manure, Microb. Ecol., 23, 239–255, doi:10.1007/BF00164099, 1992.

Zhang, J., Larsen Kolstad, E., Zhang, W., Vogeler, I. and Petersen, S. O.: Modeling coupled nitrification-denitrification in soil with an organic hotspot, Biogeosciences Discuss., doi:10.5194/bg-2023-98, 2023.